# BILEVEL OPTIMIZATION WITH LOWER-LEVEL UNIFORM CONVEXITY: THEORY AND ALGORITHM

**Yuman Wu, Xiaochuan Gong, Jie Hao, Mingrui Liu**[*]
Department of Computer Science, George Mason University
Fairfax, VA 22030, USA
{ywu45, xgong2, jhao6, mingruil}@gmu.edu

## ABSTRACT

Bilevel optimization is a hierarchical framework where an upper-level optimization problem is constrained by a lower-level problem, commonly used in machine learning applications such as hyperparameter optimization. Existing bilevel optimization methods typically assume strong convexity or Polyak-Łojasiewicz (PL) conditions for the lower-level function to establish non-asymptotic convergence to a solution with small hypergradient. However, these assumptions may not hold in practice, and recent work (Chen et al., 2024) has shown that bilevel optimization is inherently intractable for general convex lower-level functions with the goal of finding small hypergradients.

In this paper, we identify a tractable class of bilevel optimization problems that interpolates between lower-level strong convexity and general convexity via *lower-level uniform convexity*. For uniformly convex lower-level functions with exponent $p \geq 2$, we establish a novel implicit differentiation theorem characterizing the hyperobjective's smoothness property. Building on this, we design a new stochastic algorithm, termed UniBiO, with provable convergence guarantees, based on an oracle that provides stochastic gradient and Hessian-vector product information for the bilevel problems. Our algorithm achieves $\widetilde{O}(\epsilon^{-5p+6})$ oracle complexity bound for finding $\epsilon$-stationary points. Notably, our complexity bounds match the optimal rates in terms of the $\epsilon$ dependency for strongly convex lower-level functions ($p = 2$), up to logarithmic factors. Our theoretical findings are validated through experiments on synthetic tasks and data hyper-cleaning, demonstrating the effectiveness of our proposed algorithm.

## 1 INTRODUCTION

Bilevel optimization (Bracken & McGill, 1973; Dempe, 2002) is a hierarchical optimization framework where an upper-level optimization problem is constrained by a lower-level optimzation problem. Bilevel optimization plays a crucial role in various machine learning applications, including meta-learning (Finn et al., 2017), hyperparameter optimization (Franceschi et al., 2018), data hypercleaning (Franceschi et al., 2017; Shaban et al., 2019), continual learning (Borsos et al., 2020; Hao et al., 2023), neural network architecture search (Liu et al., 2018), and reinforcement learning (Konda & Tsitsiklis, 1999). The bilevel optimization problem can be defined as:

$$\min_{x \in \mathbb{R}^{d_x}} \phi(x) := f(x, y^*(x)), \qquad y^*(x) \in \arg\min_{y \in \mathbb{R}^{d_y}} g(x, y), \tag{1}$$

where $f$ and $g$ are referred to as upper-level and lower-level functions respectively. A common assumption in bilevel optimization is that the lower-level function is either strongly convex (Ghadimi & Wang, 2018; Hong et al., 2023; Ji et al., 2021; Chen et al., 2021a; 2023; Hao et al., 2024; Kwon et al., 2023a) or satisfies the Polyak-Łojasiewicz (PL) condition (Liu et al., 2022; Kwon et al., 2023b; Shen & Chen, 2023; Huang, 2024), which facilitates the design of algorithms with non-asymptotic convergence guarantees for finding a solution with a small hypergradient. However, these assumptions do not always hold in practice.

---

[*]Correspondence Author: Mingrui Liu (mingruil@gmu.edu).

Recent work (Chen et al., 2024) has explored the relaxation of these conditions but has primarily yielded negative results. Specifically, they show that for general convex lower-level problems, bilevel optimization can be intractable with the goal of finding a point with a small hypergradient: the hyperobjective function can be discontinuous and may lack stationary points. This stark contrast between lower-level strong convexity (LLSC) and mere lower-level convexity (LLC) naturally raises the following question:

**Can we identify an intermediate class of bilevel optimization problems that bridges the gap between LLSC and LLC, enabling the design of efficient algorithms of finding small hypergradients in polynomial time?**

In this paper, we provide a positive answer to this question by introducing a function class that satisfies a property called *lower-level uniform convexity* (LLUC)[1]. This property serves as a natural interpolation between LLSC and LLC, controlled by an exponent $p$. Uniform convexity (Zălinescu, 1983; Iouditski & Nesterov, 2014) is a refined notion of convexity characterized by $p \geq 2$, where $p = 2$ corresponds to strong convexity.

Finding small hypergradients under LLUC presents several challenges. First, for uniformly convex lower-level functions, the Hessian of the lower-level objective may be singular, making it impossible to compute hypergradients directly using the standard implicit differentiation theorem applicable under LLSC (Ghadimi & Wang, 2018). Second, the LLUC property inherently conflicts with the standard smoothness assumptions for the lower-level function (i.e., Lipschitz-continuous gradient in terms of the lower-level variable), which are crucial for the theoretical analysis of existing bilevel optimization algorithms (Ghadimi & Wang, 2018; Hong et al., 2023; Ji et al., 2021; Kwon et al., 2023a; Hao et al., 2024). Consequently, addressing bilevel optimization under LLUC necessitates the development of a fundamentally different algorithmic framework and novel analysis techniques.

In this work, we tackle these challenges with two key innovations. First, we develop a novel implicit differentiation theorem under LLUC, which characterizes the smoothness property of the hyperobjective, where the degree of smoothness depends on the uniformly convex exponent $p$. Second, to overcome the lack of standard smoothness assumptions for the lower-level function, we propose a new stochastic algorithm called UniBiO (Uniformly Convex Bilevel Optimization). After a warm-start stage for the lower-level variable, UniBiO employs a normalized momentum update for the upper-level variable and a multistage stochastic gradient descent with a shrinking ball strategy to update the lower-level variable. Notably, the lower-level updates are required only periodically rather than at every iteration. Our main contributions are summarized as follows.

- We identify a tractable class of bilevel optimization problems that interpolates between LLSC and LLC by leveraging the LLUC. Under this problem class, we develop a novel implicit differentiation theorem that provides an explicit hypergradient formula and establishes its smoothness property. This theorem is of independent interest and could be applied to other hierarchical optimization settings (e.g., multilevel and minimax optimization).

- We design a new stochastic algorithm named UniBiO, the first algorithm designed for bilevel optimization under LLUC. We prove that UniBiO achieves the oracle complexity $\widetilde{O}(\epsilon^{-5p+6})$ for finding an $\epsilon$-stationary point for the hyperobjective in the stochastic setting, where the oracle provides either stochastic gradients or Hessian-vector products. Notably, this oracle complexity matches the optimal complexity for strongly convex lower-level functions ($p = 2$) up to logarithmic factors.

- We conduct experiments on both an synthetic task and data hypercleaning, which validate our theory and show the effectiveness of our proposed algorithm.

## 2 RELATED WORK

**Bilevel Optimization with Lower-Level Strong Convexity.** Early research on bilevel optimization primarily focused on asymptotic convergence guarantees (Vicente et al., 1994; Anandalingam & White, 1990; White & Anandalingam, 1993). A major breakthrough came with Ghadimi & Wang (2018), which established the first non-asymptotic convergence guarantees for finding a solution

---

[1]The definition of LLUC is given in Assumption 3.2 (i).

with a small hypergradient under the assumption that the lower-level function is strongly convex. This work laid the foundation for a series of subsequent studies that improved either the complexity or the simplicity of algorithm design (Hong et al., 2023; Chen et al., 2021b; Ji et al., 2021; Kwon et al., 2023a; Hao et al., 2024; Gong et al., 2024a; Chen et al., 2021a; Khanduri et al., 2021; Dagréou et al., 2022; Guo et al., 2021; Yang et al., 2021; Gong et al., 2024b). These works critically rely on the implicit differentiation theorem from Ghadimi & Wang (2018), which is applicable under the assumption of lower-level strong convexity. In contrast, our work does not assume LLSC, rendering the standard implicit differentiation technique from Ghadimi & Wang (2018) inapplicable.

**Bilevel Optimization with Lower-Level Nonconvexity.** Bilevel optimization with nonconvex lower-level functions is generally intractable without additional assumptions (Daskalakis et al., 2021). One common approach assumes that the lower-level function satisfies the Polyak-Łojasiewicz (PL) condition (Liu et al., 2022; Kwon et al., 2023b; Shen & Chen, 2023; Huang, 2024; Chen et al., 2024). Another line of work leverages sequential approximation minimization techniques (Liu et al., 2021a;b; 2020) to solve bilevel problems without assuming lower-level strong convexity, though these methods typically offer only asymptotic convergence guarantees. Additionally, Arbel & Mairal (2022) employs Morse theory to extend implicit differentiation in the presence of multiple lower-level minima caused by nonconvexity. In contrast, our work focuses on a class of uniformly convex lower-level problems.

**Bilevel Optimization with General Lower-level Convexity.** Despite the negative results of Chen et al. (2024) under LLC from the hypergradient perspective, there is a line of work which investigates algorithms converging to $\epsilon$-KKT solution of a corresponding constrained optimization problem (Lu & Mei, 2024a;b). In contrast, our work focuses on finding an solution with small hypergradient, not an $\epsilon$-KKT solution for a corresponding constrained problem.

**Optimization for Uniformly Convex Functions.** For an single-level optimization problem under uniform convexity, the work of Iouditski & Nesterov (2014) established first-order algorithms with optimal complexity upper bounds for nonsmooth functions with bounded gradients. Under a high-order smoothness assumption, the work of Song et al. (2019) designed high-order methods for uniformly convex functions. In addition, the work of Bai & Bullins (2024) derived lower bounds for a class of optimization problems characterized by high-order smoothness and uniform convexity. In contrast, our work focuses on updating the lower-level variable using first-order methods under LLUC, without bounded gradients or smoothness assumptions.

## 3 PRELIMINARIES

Define $\|\cdot\|$ as the Euclidean norm (spectral norm) when the argument is a vector (an square matrix). Define $\langle\cdot,\cdot\rangle$ as the inner-product in Euclidean space. Denote $\odot$ by the Hadamard (element-wise) product. For any $a \in \mathbb{R}^d$, We adopt the notation $[a]^{\circ\rho} = (a_1^\rho,\dots,a_d^\rho)$ for $a \in \mathbb{R}^d$ to denote the element-wise power of a vector., where $\rho > 0$ can be any positive number (e.g., integers or non-integers). We use asymptotic notation $\widetilde{O}(\cdot), \widetilde{\Theta}(\cdot), \widetilde{\Omega}(\cdot)$ to hide polylogarithmic factors in terms of $1/\epsilon$. Define $f : \mathbb{R}^{d_x} \times \mathbb{R}^{d_y} \mapsto \mathbb{R}$ as the upper-level function, and $g : \mathbb{R}^{d_x} \times \mathbb{R}^{d_y} \mapsto \mathbb{R}$ as the lower-level function. We consider the stochastic optimization setting: we only have noisy observation of $f$ and $g$: $f(x,y) = \mathbb{E}_{\xi\sim\mathcal{D}_f}[F(x,y;\xi)]$ and $g(x,y) = \mathbb{E}_{\zeta\sim\mathcal{D}_g}[G(x,y;\zeta)]$, where $\mathcal{D}_f$ and $\mathcal{D}_g$ are underlying data distributions for upper-level function and lower-level functions respectively. We need the following definition of the differentiability in the normed vector space.

**Definition 3.1** (Differentiability in Normed Vector Spaces). Let $(X, \|\cdot\|_X)$ and $(Y, \|\cdot\|_Y)$ be normed vector spaces, let $E \subseteq X$ and $x_0 \in E$ be an accumulation point of $E$. The function $\ell : E \to Y$ is defined to be differentiable at $x_0$ if there exists a continuous linear function $J : X \to Y$ (depending on $f$ and $x_0$) such that:

$$\lim_{x\to x_0} \frac{\ell(x) - \ell(x_0) - J(x - x_0)}{\|x - x_0\|_X} = 0. \tag{2}$$

In addition, $J$ is defined as the derivative of $h$ in terms of $x$ at the point $x_0$, i.e., $J := \frac{d\ell(x)}{dx}|_{x=x_0}$.

In the following, we will introduce the problem class of LLUC with corresponding assumptions in Section 3.1, and provide some examples within the problem class in Section 3.2.

## 3.1 THE LOWER-LEVEL UNIFORM CONVEXITY PROBLEM CLASS

In this section, we introduce the assumptions that define the LLUC problem class. In particular, we identity the assumptions for both upper-level function $f$, lower-level function $g$ and the hyperobjective $\Phi$. We make the following assumptions throughout this paper.

**Assumption 3.2.** The following conditions hold for the lower-level function $g$ for some $p \geq 2$.

- (i) For every $x$, $g(x, y)$ is $(\mu, p)$-uniformly-convex with respect to $y$: $g(x, y_2) \geq g(x, y_1) + \langle \nabla_y g(x, y_1), y_2 - y_1 \rangle + \frac{\mu}{p} \|y_2 - y_1\|^p$ holds for any $y_1, y_2$.

- (ii) $g(x, y)$ is $(L_0, L_1)$-smooth in $y$ for any given $x$: $\|\nabla_{yy}^2 g(x, y)\| \leq L_0 + L_1 \|\nabla_y g(x, y)\|$ for any $y$ and any $x$.

- (iii) $\nabla_y g(x, y)$ is $l_{g,1}$-Lipschitz in $x$: $\|\nabla_y g(z_1) - \nabla_y g(z_2)\| \leq l_{g,1} \|x_1 - x_2\|$ for any $z_1 = (x_1, y), z_2 = (x_2, y) \in \mathbb{R}^{d_x + d_y}$.

- (iv) $\nabla_{xy}^2 g(x, y)$ is $l_{g,2}$-Lipschitz jointly in $(x, y)$: $\|\nabla_{xy}^2 g(z_1) - \nabla_{xy}^2 g(z_2)\| \leq l_{g,2} \|z_1 - z_2\|$ for any $z_1 = (x_1, y_1), z_2 = (x_2, y_2) \in \mathbb{R}^{d_x + d_y}$.

- (v) $\frac{d\nabla_y g(x,y)}{d[y]^{\circ p-1}}$ exists ($\frac{d\nabla_y g(x,y)}{d[y]^{\circ p-1}}$ is defined in definition A.1) and $l_{g,2}$ jointly Lipschitz continuous with $(x, y)$: $\left\| \frac{d\nabla_y g(x_1,y_1)}{d[y_1]^{\circ p-1}} - \frac{d\nabla_y g(x_2,y_2)}{d[y_2]^{\circ p-1}} \right\| \leq l_{g,2} \|z_1 - z_2\|$ holds for any $z_1 = (x_1, y_1), z_2 = (x_2, y_2) \in \mathbb{R}^{d_x + d_y}$, where $\|\frac{d\nabla_y g(x,y)}{d[y]^{\circ p-1}}\| := \sup_{\|z\|=1, z\in\mathbb{R}^{d_y}} \|\frac{d\nabla_y g(x,y)}{d[y]^{\circ p-1}} z\|$. We assume that the generalized Jacobian satisfies $\lambda_{\min} \left( \frac{d\nabla_y g(x,y)}{d[y]^{\circ(p-1)}} \right) \geq \mu > 0$.

- (vi) $\|\frac{d\nabla_y g(x,y)}{d[y]^{\circ p-1}}\| \leq C$ for some $C > 0$.

**Remark**: Assumption 3.2 specifies the key conditions imposed on the lower-level function. In particular: (i) establishes uniform convexity (Zălinescu, 1983; Iouditski & Nesterov, 2014), a generalization of strong convexity that offers greater flexibility. (ii) introduces a relaxed smoothness condition (Zhang et al., 2020), which differs from the standard $L$-smooth assumption. The standard $L$-smooth condition is incompatible with uniform convexity when the domain is unbounded, making this relaxation more appropriate. (iii) and (iv) are standard assumptions commonly adopted in bilevel optimization (Ghadimi & Wang, 2018; Hong et al., 2023; Ji et al., 2021; Kwon et al., 2023a). (v) and (vi) impose differentiability of $\nabla_y g(x, y)$ with respect to $[y]^{\circ p-1}$ (as defined in definition 3.1, with the complete definition in definition A.1). These two conditions are essential for developing the implicit differentiation theorem under LLUC in Section 4. Note that the assumption (v) can be replaced by the assumption that $\frac{d\nabla_y g(x,y)}{d[y]^{\circ p-1}}$ is independent of $[y]^{\circ p-1}$, and more details can be found in Appendix B.2. When $p = 2$, the uniformly convex function becomes strongly convex, the generalized Hessian becomes the standard Hessian matrix $\nabla_{yy} g(x, y)$, which is positive definite.

**Assumption 3.3.** The following conditions hold for the upper-level function $f$ for some $p \geq 2$:

- (i) $\nabla_x f(x, y)$ is $l_{f,1}$-jointly Lipschitz in $(x, y)$: $\|\nabla_x f(z_1) - \nabla_x f(z_2)\| \leq l_{f,1} \|z_1 - z_2\|$ for any $z_1 = (x_1, y_1), z_2 = (x_2, y_2) \in \mathbb{R}^{d_x + d_y}$;

- (ii) $\frac{df(x,y)}{d[y]^{\circ p-1}}$ exists and $l_{f,1}$-jointly Lipschitz in $(x, y)$: $\|\frac{df(x_1,y_1)}{d[y_1]^{\circ p-1}} - \frac{df(x_2,y_2)}{d[y_2]^{\circ p-1}}\| \leq l_{f,1} \|z_1 - z_2\|$ for any $z_1 = (x_1, y_1) \in \mathbb{R}^{d_x + d_y}, z_2 = (x_2, y_2) \in \mathbb{R}^{d_x + d_y}$;

- (iii) $\|\frac{df(x,y)}{d[y]^{\circ p-1}}\| \leq l_{f,0}$ for any $x \in \mathbb{R}^{d_x}$ and any $y \in \mathbb{R}^{d_y}$.

- (iv) There exists $\Delta_\phi \geq 0$ such that $\Phi(x_0) - \inf_x \Phi(x) \leq \Delta_\phi$.

**Remark 1**: Assumption 3.3 characterizes the assumptions we need for the upper-level function $f$ and the hyperobjective $\Phi$. In particular: (i) and (iv) are standard assumptions in the nonconvex and bilevel optimization literature (Ghadimi & Lan, 2013; Ghadimi & Wang, 2018; Hong et al., 2023; Ji et al., 2021; Kwon et al., 2023a). (ii) and (iii) impose differentiability of $f(x, y)$ in terms of $[y]^{\circ p-1}$ (as defined in Definition A.1), which is satisfied for a class of functions satisfying growth condition (See

Appendix B.7 for more details). These two conditions are also crucial for the implicit differentiation theorem under LLUC in Section 4.

**Remark 2**: If the differentiability assumption in Assumption 3.2 (v) (vi) and Assumption 3.3 (ii) (iii) hold with respect to the variable $[y-a]^{\circ p-1}$ with some vector $a \in \mathbb{R}^{d_y}$, the analysis of the implicit differentiation theorem in Section 4 is the same as in the case of $a = 0$. Without loss of generality, we simply assume $a = 0$ for the clean presentation. More details are illustrated in Appendix B.4.

**Assumption 3.4.** We access stochastic estimators through an unbiased oracle and they satisfy:

$$\mathbb{E}_{\xi \sim \mathcal{D}_f}[\|\nabla_x F(x, y; \xi) - \nabla_x f(x, y)\|^2] \leq \sigma_f^2,$$

$$\mathbb{E}_{\zeta \sim \mathcal{D}_g}[\exp(\|\nabla_y G(x, y; \zeta) - \nabla_y g(x, y)\|^2 / \sigma_{g,1}^2)] \leq \exp(1),$$

$$\mathbb{E}_{\zeta \sim \mathcal{D}_g}[\|\nabla_{xy} G(x, y; \zeta) - \nabla_{xy} g(x, y)\|^2] \leq \sigma_{g,2}^2,$$

$$\mathbb{E}_{\xi \sim \mathcal{D}_f}\left[\left\|\frac{dF(x, y; \xi)}{d[y]^{\circ p-1}} - \frac{df(x, y)}{d[y]^{\circ p-1}}\right\|^2\right] \leq \sigma_f^2,$$

$$\mathbb{E}_{\zeta \sim \mathcal{D}_g}\left[\left\|\frac{d\nabla_y G(x, y; \zeta)}{d[y]^{\circ p-1}} - \frac{d\nabla_y g(x, y)}{d[y]^{\circ p-1}}\right\|^2\right] \leq \sigma_{g,2}^2. \tag{3}$$

**Remark**: Theorem 3.4 states that the stochastic oracle has bounded variance, which is a standard assumption in nonconvex stochastic optimization (Ghadimi & Lan, 2013; Ghadimi & Wang, 2018; Ji et al., 2021). Additionally, it assumes that the stochastic first-order oracle for the lower-level problem is light-tailed, a common requirement for high-probability analysis in lower-level optimization (Lan, 2012; Hazan & Kale, 2014; Hao et al., 2024; Gong et al., 2024a). Our unique assumptions under LLUC are presented in Eq. (3), assuming bounded variance for generalized derivative and generalized Hessian for upper-level and lower-level functions. When $p = 2$, these assumptions recover the standard ones in bilevel optimization under LLSC (Ghadimi & Wang, 2018; Hong et al., 2023).

We use Neumann series approach (Ghadimi & Wang, 2018; Ji et al., 2021) to approximate the hypergradient. Define

$$\hat{\nabla} f(x, y; \bar{\xi}) := \nabla_x F(x, y; \xi)$$

$$- \nabla_{xy} G(x, y; \zeta^{(0)}) \left[\frac{1}{C} \sum_{q=0}^{Q-1} \prod_{j=1}^{q} \left(I - \frac{1}{C} \frac{d\nabla_y G(x, y; \zeta^{(q,j)})}{d[y]^{\circ p-1}}\right)\right] \frac{dF(x, y; \xi)}{d[y]^{\circ p-1}}, \tag{4}$$

where $\hat{\nabla} f(x, y; \bar{\xi})$ is the stochastic approximation of hypergradient $\nabla \Phi(x)$ and the randomness $\bar{\xi}$ is defined as $\bar{\xi} := \{\xi, \zeta^{(0)}, \bar{\zeta}^{(0)}, \ldots, \bar{\zeta}^{(Q-1)}\}$ with $\bar{\zeta}^{(q)} := \{\zeta^{(q,1)}, \ldots, \zeta^{(q,q)}\}$.

## 3.2 EXAMPLES

In this section, we provide two examples of bilevel optimization problems where the lower-level problem is uniformly convex. More examples can be found in Appendix A.2.

**Example 1.** $f(x, y) = y^3$, $g(x, y) = \frac{1}{4} y^4 - y \sin x$. In this example, the LLUC holds with $p = 4$.

**Example 2 (Data Hypercleaning).** The data hypercleaning task (Shaban et al., 2019) aims to learn a set of weights $\lambda$ to the noisy training dataset $\mathcal{D}_{tr}$, such that training a model on the weighted training set can leads to a strong performance on the clean validation set $\mathcal{D}_{val}$. The noisy set is defined as $\mathcal{D}_{tr} := \{x_i, \bar{y}_i\}$, where each label $\bar{y}_i$ is independently flipped to a different class with probability $0 < \tilde{p} < 1$. This problem can be formulated as a bilevel optimization task:

$$\min_{\lambda} \frac{1}{|\mathcal{D}_{\text{val}}|} \sum_{\xi \in \mathcal{D}_{\text{val}}} \mathcal{L}(w^*(\lambda); \xi),$$

$$\text{s.t.} \quad w^*(\lambda) \in \arg\min_{w} \frac{1}{|\mathcal{D}_{\text{tr}}|} \sum_{\zeta_i \in \mathcal{D}_{\text{tr}}} \sigma(\lambda_i) \mathcal{L}(w; \zeta_i) + c\|w\|_p^p, \tag{5}$$

where $w$ represents the model parameters, and $\sigma(x) = \frac{1}{1+e^{-x}}$ is the sigmoid function. Note that the LLUC condition holds when the lower-level problem is a $\ell_p$ norm regression (Woodruff & Zhang,

2013; Jambulapati et al., 2022) problem for $p \geq 2$, with/without a uniformly convex regularizer $\|w\|_p^p$ (Sridharan & Tewari, 2010).

If we choose $\mathcal{L}(w; \xi)$ in Equation (5) to be $\mathcal{L}(w; \zeta_i) = |x_i^\top w - \bar{y}_i|^p$, where $\zeta_i = (x_i, \bar{y}_i)$ is the $i$-th training sample. In this case, the lower-level problem in Equation (5) becomes

$$g(w, \lambda) = \frac{1}{n} \|\Lambda(Xw - \bar{y})\|_p^p + c\|w\|_p^p, \quad \Lambda = \mathrm{diag}(\sigma(\lambda_1)^{1/p}, \ldots, \sigma(\lambda_n)^{1/p}), \tag{6}$$

$$X = [x_1^\top; \ldots; x_n^\top] \in \mathbb{R}^{n \times d}, \quad \bar{y} = [\bar{y}_1, \ldots, \bar{y}_n]^\top \in \mathbb{R}^{n \times 1}, \quad w \in \mathbb{R}^d.$$

We know that $g(w, \lambda)$ is a sum of two uniformly convex functions, and hence is uniformly convex by Assumption 3.2 (i): the summation of a $(\mu_1, p)$ and $(\mu_2, p)$-uniformly-convex functions is $(\mu_1 + \mu_2, p)$-uniformly-convex. The specific value of $\mu_1$ and $\mu_2$ can be found in Appendix A.2.

The detailed proof is included in Appendix A.2. The key characteristic is that the lower-level function $g$ is not a strongly convex function in terms of $y$ when $p > 2$.

## 4 Implicit Differentiation Theorem under LLUC

In this section, we present the implicit differentiation theorem under the LLUC condition. A key technical challenge arises from the singular Hessian of the lower-level function, which renders the standard implicit function theorem (Ghadimi & Wang, 2018) inapplicable in our setting. To overcome this, our theorem explicitly exploits the uniform convexity of the lower-level function and its high-order differentiability to establish the differentiability of the hyperobjective, along with its smoothness property. The formal statement is given in Theorem 4.1.

**Theorem 4.1** (Implicit Differentiation Theorem under LLUC). *Suppose Assumption 3.2 and 3.3 hold. Then $\Phi$ is differentiable in $x$ and can be computed as the following:*

$$\nabla\Phi(x) = \nabla_x f(x, y^*(x)) - \nabla_{xy} g(x, y^*(x)) \left[ \frac{d\nabla_y g(x, y^*(x))}{d[y^*(x)]^{\circ p-1}} \right]^{-1} \frac{df(x, y^*(x))}{d[y^*(x)]^{\circ p-1}}. \tag{7}$$

*In addition, the function $\Phi$ satisfies the following properties:*

$$\|\nabla\Phi(x_1) - \nabla\Phi(x_2)\| \leq L_{\phi_1} \|x_1 - x_2\|^{\frac{1}{p-1}} + L_{\phi_2} \|x_1 - x_2\|, \tag{8}$$

$$\Phi(x_1) \leq \Phi(x_2) + \langle \nabla\Phi(x_2), x_1 - x_2 \rangle + \frac{(p-1)L_{\phi_1}}{p} \|x_1 - x_2\|^{\frac{p}{p-1}} + \frac{L_{\phi_2}}{2} \|x_1 - x_2\|^2. \tag{9}$$

*where $l_p = \left( \frac{pl_{g,1}}{\mu} \right)^{\frac{1}{p-1}}$, $L_{\phi_1} = l_p(l_{f,1} + \frac{l_{f,2}l_{g,2}}{\mu} + \frac{l_{g,1}l_{f,1}}{\mu} + \frac{l_{g,1}l_{f,1}l_{g,2}}{\mu^2})$, $L_{\phi_2} = l_{f,1} + \frac{l_{f,2}l_{g,2}}{\mu} + \frac{l_{g,1}l_{f,1}}{\mu} + \frac{l_{g,1}l_{f,1}l_{g,2}}{\mu^2}$.*

**Remark**: Theorem 4.1 provides an explicit formula Eq. (7) to calculate the hypergradient, as well as the smoothness property of $\Phi$ characterized in Eq. (8). In addition, it includes the descent inequality Eq. (9), which plays a crucial role in the algorithmic analysis under LLUC in Section 5. Notably, when $p = 2$, this theorem recovers the standard implicit function theorem under LLSC (Ghadimi & Wang, 2018). Intuitively, as $p$ increases, the lower-level function deviates further from strong convexity, and hence the smoothness property of the hyperobjective becomes worse. The proof of Theorem 4.1 is included in Appendix B.3.

### 4.1 Proof Sketch

In this section, we provide a proof sketch for the proof of Theorem 4.1. The key idea is to prove two things under Assumptions 3.2 and 3.3: (1) the optimal lower-level variable is Hölder continuous in terms of upper-level variable, which is stated in Lemma 4.2; (2) the generalized Hessian after the change of variable (i.e., $y$ is replaced to $[y]^{\circ p-1}$) has a positive minimum eigenvalue and hence is invertible, which is stated in Lemma B.2. These two lemmas can be regarded as counterparts of the implicit differentiation theorem under LLSC (Ghadimi & Wang, 2018).

**Lemma 4.2** (Hölder Continuity of the Lower-Level Optimal Solution Mapping). *$y^*(x)$ is hölder continuous: for any $x_1, x_2 \in \mathbb{R}^{d_x}$, we have $\|y^*(x_2) - y^*(x_1)\| \leq l_p \|x_2 - x_1\|^{\frac{1}{p-1}}$, where $l_p$ is defined in Theorem 4.1.*

---

**Algorithm 1** EPOCH-SGD

---

1: **Input:** function $\psi$, $\gamma_1$, $T_1$, $D_1$, and total time $T$
2: **Initialize:** $w_1^1$, set $\tau = 2(p-1)/p$ and $k = 1$
3: **while** $\sum_{i=1}^{k} T_i \leq T$ **do**
4:    **for** $t = 1, \ldots, T_k$ **do**
5:       $w_{t+1}^k = \Pi_{w \in \mathcal{B}(w_1^k, D_k)}(w_t^k - \gamma_k \nabla \psi(w_t^k; \pi_t^k))$
6:    **end for**
7:    $w_1^{k+1} = \frac{1}{T_k} \sum_{t=1}^{T_k} w_t^k$
8:    $T_{k+1} = 2^\tau T_k$, $\gamma_{k+1} = \gamma_k/2$, $D_{k+1} = D_k/2^{\frac{1}{p}}$.
9:    $k \leftarrow k + 1$
10: **end while**
11: **Return** $w_1^k$

---

**Remark:** This lemma shows that the optimal lower-level variable $y^*(x)$ is Hölder continuous in terms of the upper-leval variable $x$, with the exponent $\frac{1}{p-1}$. When $p = 2$, this lemma recovers the standard Lipschitz continuous condition of $y^*(x)$ under LLSC (Ghadimi & Wang, 2018). It is worth nothing that the existing bilevel optimization algorithms with nonasymptotic convergence guarantees to $\epsilon$-stationary point all require the Lipschitzness of $y^*(x)$ (Ghadimi & Wang, 2018; Hong et al., 2023; Ji et al., 2021; Kwon et al., 2023b; Chen et al., 2024).

Building on Lemma 4.2, we are ready to show the hyperobjective is differentiable everywhere and establish the smoothness property of the hyperobjective. The detailed proof of Theorem 4.1 is included in Appendix B.3.

## 5 ALGORITHM AND CONVERGENCE ANALYSIS

### 5.1 ALGORITHM DESIGN

In this section, we introduce our algorithm design techniques, leveraging our implicit differentiation theorem under LLUC. A natural approach is as follows: for a fixed upper-level variable $x$, one can iteratively update the lower-level variable until it sufficiently approximates $y^*(x)$, ensuring an accurate hypergradient estimation. The upper-level variable $x$ can then be updated accordingly. However, this naive method may suffer from a high oracle complexity. To design an algorithm with better oracle complexity, our algorithm updates the upper-level variable by normalized momentum, while the lower-level variable is updated by an variant of Epoch-SGD (Hazan & Kale, 2014) periodically. The algorithm is similar to the BO-REP algorithm in Hao et al. (2024), but with a crucial distinction: while BO-REP is designed for strongly convex lower-level problems and relaxed smooth hyperobjectives, our UniBiO algorithm is tailored for uniformly convex and relaxed smooth lower-level problems with Hölder-smooth hyperobjectives. Therefore, despite conceptual similarities in the update mechanism, UniBiO requires significantly different hyperparameter choices, such as the learning rate, periodic update intervals, and the number of iterations.

The detailed description of our algorithm is illustrated in Algorithm 2. The algorithm starts from a warm-start stage, where the lower-level variable is updated by the epoch-SGD algorithm for a certain number of iterations under the fixed upper-level variable $x_0$ (line 3). After that, the algorithm follows a periodic update scheme for the lower-level variable, performing an update every $I$ iterations (line $4 \sim 6$), while the upper-level variable is updated at each iteration using a normalized stochastic gradient with momentum (lines $7 \sim 8$). For the lower-level update, our method employs a variant of Epoch-SGD (described in Algorithm 1), which integrates stochastic gradient descent updates with a shrinking ball strategy.

### 5.2 MAIN RESULTS

Before presenting the main result, we first introduce a few notations. Denote $\sigma(\cdot)$ as the $\sigma$-algebra generated by the random variables in the arguments. Define $\mathcal{F}_t := \sigma(\bar{\xi}_1, \ldots, \bar{\xi}_{t-1})$ for $t \geq 1$, let $\mathcal{F}_y$ be the filtration used to update $\{y_t\}_{t=0}^T$. We use $C_1$ to denote large enough constant.

---

**Algorithm 2** UNIBIO

1: **Input:** $\eta, \beta, \{\alpha_{t,1}\}, \{K_{t,1}\}, \{R_{t,1}\}, \{K_t\}, T$
2: **Initialize:** $x_1, y_0, m_{-1} = 0$
3: $y_1 = \text{EPOCH-SGD}(g(x_0, \cdot), \alpha_{0,1}, K_{0,1}, R_{0,1}, K_0)$
4: **for** $t = 1, \ldots, T$ **do**
5:     **if** $t$ is a multiple of $I$ **then**
6:         $y_t = \text{EPOCH-SGD}(g(x_t, \cdot), \alpha_{t,1}, K_{t,1}, R_{t,1}, K_t)$
7:     **end if**
8:     $m_t = \beta m_{t-1} + (1 - \beta)\hat{\nabla} f(x_t, y_t; \bar{\xi}_t)$, where $\hat{\nabla} f(x, y; \bar{\xi})$ is defined in Eq. (4)
9:     $x_{t+1} = x_t - \eta \frac{m_t}{\|m_t\|}$
10: **end for**

---

**Theorem 5.1.** *Under Assumptions 3.2, 3.3  3.4 , for any given $\delta \in (0, 1)$ and $\epsilon > 0$, we choose $\alpha_{t,1} = O(1)$, $K_{t,1} = O(1)$, $R_{t,1} = O(1)$, $K_t = \widetilde{O}(\epsilon^{-2p+2})$, $I = O(\epsilon^{-2})$, $Q = \widetilde{O}(1)$, $1 - \beta = \Theta(\epsilon^2)$, and $\eta = \Theta(\epsilon^{3p-3})$ (see Theorem D.1 for exact choices). Let $T = \frac{C_1 \Delta_\phi}{\eta \epsilon}$. Then with probability at least $1 - \delta$ over the randomness in $\mathcal{F}_y$, we have $\frac{1}{T}\sum_{t=1}^{T} \mathbb{E}\|\nabla\Phi(x_t)\| \leq \epsilon$, where the expectation is taken over the randomness in $\mathcal{F}_{T+1}$. The total oracle complexity is $\widetilde{O}(\epsilon^{-5p+6})$.*

**Remark:** The full statement of Theorem 5.1 is included in Section D. Theorem 5.1 shows that our algorithm UniBiO requires $\widetilde{O}(\epsilon^{-5p+6})$ oracle complexity for finding an $\epsilon$-stationary point. To the best of our knowledge, this is the first nonasymptotic result under LLUC. In addition, when the lower function is strongly convex ($p = 2$), the complexity bound becomes $\widetilde{O}(\epsilon^{-4})$, which matches the optimal rate in terms of the $\epsilon$ dependency (Arjevani et al., 2023) for stochastic bilevel optimization under LLSC (Dagréou et al., 2022; Chen et al., 2023). It remains unclear whether the complexity result in terms of $\epsilon$ is tight for $p > 2$.

### 5.3 PROOF SKETCH

In this section, we present a sketch of the proof for Theorem 5.1. The complete proof can be found in Appendix D. The key idea of the proof resembles the proof of Hao et al. (2024), but our proof is under a different problem setting (i.e., Hölder smooth hyperobjective and uniformly convex lower-level function). Define $y_t^* = y^*(x_t)$. Note that Algorithm 2 uses normalized momentum update, therefore $\|x_{t+1} - x_t\| = \eta$. By the Hölder continuity of $y^*(x)$ (guaranteed by Lemma 4.2), we know that $\|y_{t+1}^* - y_t^*\| \leq l_p \eta^{\frac{1}{p-1}}$. Therefore the optimal lower-level variable moves slowly across iterations when $\eta$ is small. Hence, the periodic update for the lower-level variable can still be a good estimate for the optimal lower-level variable if the length of the period $I$ is not too large. Lemma 5.2 and 5.3 are devoted to control the lower-level error, while Lemma 5.4 is devoted to control the cummulative hypergradient bias over time. Given these lemmas, one can leverage the descent inequality Eq. (9) developed in Theorem 4.1 to establish the convergence rate. The following lemmas are based on Theorems 3.2 to 3.4. The detailed proofs of this section can be found in Section C.

**Lemma 5.2.** *Under the same parameter setting as in Theorem 5.1, for any sequence $\{\tilde{x}_t\}$ such that $\tilde{x}_0 = x_0$ and $\|\tilde{x}_{t+1} - \tilde{x}_t\| = \eta$, let $\{\tilde{y}_t\}$ be the output produced by Algorithm 2 with input $\{\tilde{x}_t\}$. Then with probability at least $1 - \delta$, for all $t \in [T]$ we have $\|\tilde{y}_t - \tilde{y}_t^*\| \leq \min\{\epsilon/4L_{\phi_2}, 1/L_1\}$.*

**Remark**: Lemma 5.2 establishes a bound on the lower-level tracking error for *any* slowly varying sequence $\{\tilde{x}_t\}$ under LLUC. A key advantage of this result is that it provides lower-level guarantees independently of the randomness in the upper-level variables, avoiding potential randomness dependency issues. Similar techniques have been employed in Hao et al. (2024). The main difficulty of the proof comes from a high probability analysis for handling the convergence analysis of epoch-SGD for the lower-level variable under lower-level uniform convexity and relaxed smoothness. The complete proof of Lemma 5.2 can be found in the proof of Lemma C.9 in the Appendix.

**Corollary 5.3.** *Under the same setting as in Theorem 5.1, let $\{x_t\}$ and $\{y_t\}$ be the iterates generated by Algorithm 2. Then with probability at least $1 - \delta$ (denote this event as $\mathcal{E}$) we have $\|y_t - y_t^*\| \leq \min\{\epsilon/4L_{\phi_2}, 1/L_1\}$ for all $t \geq 1$.*

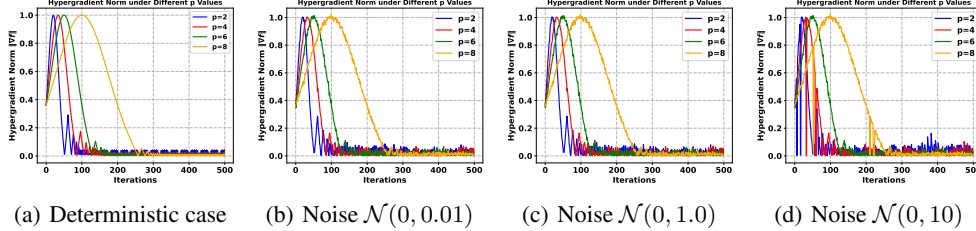

|  (a) Deterministic case | (b) Noise $\mathcal{N}(0, 0.01)$ | (c) Noise $\mathcal{N}(0, 1.0)$ | (d) Noise $\mathcal{N}(0, 10)$ |

Figure 1: Convergence results for synthetic experiments on upper-level non-convex, lower-level uniform-convex bilevel optimization with varying uniform-convex parameter $p = [2, 4, 6, 8]$ in the deterministic case and stochastic case with different types of Gaussian noise $\mathcal{N}(0, 0.01), \mathcal{N}(0, 1.0), \mathcal{N}(0, 10)$ respectively.

**Remark**: Corollary 5.3 is a direct application of Lemma 5.2. We replace the any sequence $\{\tilde{x}_t\}$ to the actual sequence $x_t$ in the Algorithm 2 and obtains the same bound. The reason is that the actual sequence in Algorithm 2 satisfies the condition in Lemma 5.2.

**Lemma 5.4.** *Define* $\epsilon_t := m_t - \nabla\Phi(x_t)$. *Under event* $\mathcal{E}$, *we have* $\sum_{t=1}^{T} \mathbb{E}\|\epsilon_t\| \leq \frac{\sigma_1}{1-\beta} + T\sqrt{1-\beta}\sigma_1 + \frac{T\epsilon}{4} + \frac{Tl_{g,1}l_{f,0}}{\mu}\left(1 - \frac{\mu}{C}\right)^Q + \frac{T}{1-\beta}\left(L_{\phi_1}\eta^{\frac{1}{p-1}} + L_{\phi_2}\eta\right)$.

**Remark**: Lemma 5.4 characterizes the cumulative bias of the hypergradient over time. When $1 - \beta$ is small (e.g., $\Theta(\epsilon^2)$ in Theorem 5.1) and $\eta$ is small (e.g., $\eta = \Theta(\epsilon^{3p-3})$), the cummulative bias grow with a sublinear rate in terms of $T$. This lemma can be regarded as a generalization of the analysis of normalized momentum for smooth functions (Cutkosky & Mehta, 2020) to bilevel problems with Hölder-smooth functions.

## 6 EXPERIMENTS

**Synthetic Experiment.** We consider the following synthetic experiment in the bilevel optimization problem illustrated in Example 3 in Appendix A: $g(x, y) = \frac{1}{p}y^p - y\sin x$, and $f(x, y) = \mathbf{1}\left(y > (\frac{\pi}{2})^{\frac{1}{p-1}}\right) - \mathbf{1}\left(y < -(\frac{\pi}{2})^{\frac{1}{p-1}}\right) + \sin(y^{p-1})\mathbf{1}\left(|y| \leq (\frac{\pi}{2})^{\frac{1}{p-1}}\right)$, where $\mathbf{1}(\cdot)$ is the indicator function, $p \geq 2$ is an even number. The goal of this experiment is to verify the complexity results established in Theorem 5.1. In theory, we expect that larger $p$ will make our algorithm UniBiO converge slower.

We conduct our experiments by implementing our proposed algorithms with varying values of $p = [2, 4, 6, 8]$. The number of upper-level iterations is fixed at $T = 500$, while the number of lower-level iterations is set to 100. To consider the effects of stochastic gradients, we introduce Gaussian noise with different variances on the gradients, specifically $\mathcal{N}(0, 10)$, $\mathcal{N}(0, 1)$, and $\mathcal{N}(0, 0.01)$. Other fixed parameters are set as $\beta = 0.9$, $I = 2$, $T_1 = 5$, and $D_1 = 1$, with initialization at the point $(x_0, y_0) = (1, 1)$. We tune the learning rates from $(0.01, 0.1)$ for both upper-level and lower-level for every $p \in [2, 4, 6, 8]$. The best learning rate choices for upper-level variable are $\eta = [0.05, 0.03, 0.02, 0.01]$ for $p = [2, 4, 6, 8]$, respectively, while the best lower-level learning rate for every $p$ is $\alpha = [1, 1, 1, 1]$ corresponding to $p = [2, 4, 6, 8]$.

Figure 1 presents the results for the deterministic setting (a) and the stochastic settings (b) (c) (d) with Gaussian noise with variances 0.01, 1 and 10 respectively. Our experimental results empirically validate the theoretical analysis of our algorithm, demonstrating that an increase in the lower-level parameter $p$ leads to a deterioration in computational complexity. This observation aligns with our theoretical results. Additional experiments for various values of $p$ and other bilevel optimization baselines (such as StocBiO (Ji et al., 2021), TTSA (Hong et al., 2023) and MA-SOBA (Chen et al., 2023)) are included in Appendix E.1.

**Data Hypercleaning.** To verify the effectiveness of the proposed UniBiO algorithm, we conduct data hypercleaning experiments (Shaban et al., 2019) and compare with other baselines as formulated

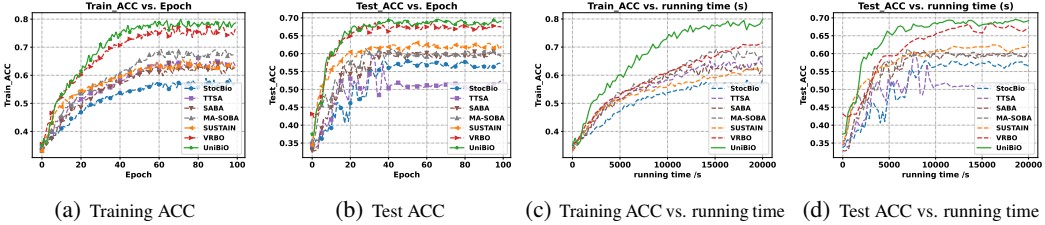

(a) Training ACC      (b) Test ACC      (c) Training ACC vs. running time      (d) Test ACC vs. running time

Figure 2: Results of bilevel optimization on data hyper-cleaning with probability $\tilde{p} = 0.1$ and the uniformly convex regularizer $\|w\|_p^p$ with $p = 3$. Subfigure (a), (b) show the training and test accuracy with the training epoch. Subfigures (c), (d) show the training and test accuracy with the running time.

in Eq. (5). To evaluate this approach, we apply our proposed bilevel algorithms and other baselines to a noisy version of the Stanford Natural Language Inference (SNLI) dataset (Bowman et al., 2015) (under Creative Commons Attribution-ShareAlike 4.0 International License), a text classification task. The model used is a three-layer recurrent neural network with an input dimension of 300, a hidden dimension of 4096, and an output dimension of 3, predicting labels among entailment, contradiction, and neutral. In our experiment, each training sample's label is randomly altered to one of the other two categories with probability $0.1$. All the experiments are run on an single NVIDIA A6000 (48GB memory) GPU and a AMD EPYC 7513 32-Core CPU. We have also included the experiment of $p = 4$ in Appendix E.2. Our method achieves higher classification accuracy on both the training and test sets compared with baselines, as illustrated in Figure 4. Moreover, it demonstrates strong computational efficiency. Further details on parameter selection and tuning are provided in Appendix F. The code is available at `https://github.com/MingruiLiu-ML-Lab/bilevel-optimization-lower-level-uniform-convexity`.

## 7 CONCLUSION

In this paper, we identify a tractable class of bilevel optimization problems that interpolates between lower-level strong convexity and general convexity via lower-level uniform convexity. We develop a novel implicit differentiation theorem under LLUC characterizing the hyperobjective's smoothness property. Based on this, we introduce UniBiO, a new stochastic algorithm that achieves $\widetilde{O}(\epsilon^{-5p+6})$ oracle complexity for finding $\epsilon$-stationary points. Experiments on an synthetic task and a data hyper-cleaning task demonstrate the superiority of our proposed algorithm. One limitation is that our algorithm design requires the prior knowledge of $p$, but in practice, such a knowledge of $p$ may not be available. Designing a universal bilevel optimization algorithm that adapts to $p$ without explicit knowledge in the spirit of Nesterov (2015) is an important challenge.

## REPRODUCIBILITY STATEMENT

We provide Theorems 4.1 and 5.1 in main text, the proof of Theorem 4.1 in Section B.3, and the proof of Theorem 5.1 in Section D. An anonymized code archive with training/evaluation scripts, configurations, seeds, and environment files is included in the supplementary materials. The dataset SNLI is accessible on HuggingFace under Creative Commons Attribution-ShareAlike 4.0 International License. We include preprocessing/splitting scripts, and references to their dataset cards and licenses. These materials sufficiently support the reproduction of our results.

## ACKNOWLEDGEMENTS

We would like to thank the anonymous reviewers for their helpful comments. This work has been supported by the Presidential Scholarship, and the IDIA P3 fellowship from George Mason University, and NSF award #2436217, #2425687. The Computations were run on Hopper, a research computing cluster provided by the Office of Research Computing at George Mason University (URL: https://orc.gmu.edu).

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

## A    PROOFS IN SECTION 3

### A.1    DEFINITION

**Definition A.1.** $\frac{df(x,y)}{d[y]^{\circ p-1}}$ and $\frac{d\nabla_y g(x,y)}{d[y]^{\circ p-1}}$ are defined as the following: for any $y$, define $z = [y]^{\circ p-1}$ and $f(x, z^{\circ \frac{1}{p-1}}), \nabla_y g(x, z^{\circ \frac{1}{p-1}})$ is differentiable with $z$. Mathematically, there exist linear mappings $J_1, J_2$ such that for any $z \in \mathbb{R}^{d_y}$, vector $h \in \mathbb{R}^{d_y}$ and any small constant $\delta$, the following statements hold:

$$
\begin{aligned}
\lim_{\delta \to 0} \frac{f(x, [z+\delta h]^{\circ \frac{1}{p-1}}) - f(x, z^{\circ \frac{1}{p-1}}) - \langle J_1, \delta h \rangle}{\|\delta h\|} &= 0, \\
\lim_{\delta \to 0} \frac{\nabla_y g(x, [z+\delta h]^{\circ \frac{1}{p-1}}) - \nabla_y g(x, z^{\circ \frac{1}{p-1}}) - J_2 \delta h}{\|\delta h\|} &= 0
\end{aligned}
\tag{10}
$$

In addition, we define $J_1 = \frac{df(x,y)}{d[y]^{\circ p-1}} = \frac{df(x, z^{\circ \frac{1}{p-1}})}{dz}$, and $J_2 = \frac{d\nabla_y g(x,y)}{d[y]^{\circ p-1}} = \frac{d\nabla_y g(x, z^{\circ \frac{1}{p-1}})}{dz}$.

### A.2    EXAMPLES

**Example 1.**    Let functions $f$ and $g$ be defined as:

$$
f(x, y) = y^3, \quad g(x, y) = \frac{1}{4}y^4 - y \sin x. \tag{11}
$$

Now we verify the assumptions.

- Assumption 3.2 (i): Since $\frac{1}{4}y^4$ is a $(1, 4)$-uniform convex function , $y \sin x$ is a linear function with $y$, so $g(x, y) = \frac{1}{4}y^4 - y \sin x$ is $(1, 4)$ uniform convex with $y$.

- Assumption 3.2 (ii): $\|\nabla_{yy} g(x, y)\| = 3y^2 \leq 12 + 6\|y^3 - \sin x\| = 12 + 6\|\nabla_y g(x, y)\|$, hence we have $L_0 = 12, L_1 = 6$.

- Assumption 3.2 (iii): $\nabla_y g(x, y) = y^3 - \sin x$, so $\|\nabla_y g(x_1, y) - \nabla_y g(x_2, y)\| = \|\sin x_2 - \sin x_1\| \leq \|x_1 - x_2\|$. Therefore $l_{g,1} = 1$.

- Assumption 3.2 (iv): $\nabla_{xy} g(x, y) = -\cos x$, so $\|\nabla_{xy} g(x_1, y_1) - \nabla_{xy} g(x_2, y_2)\| = \|\cos x_2 - \cos x_1\| \leq \|x_1 - x_2\|$. Therefore $l_{g,2} = 1$.

- Assumption 3.2 (v): $\nabla_y g(x, y) = y^3 - \sin x$, so $\frac{d\nabla_y g(x,y)}{d[y]^{\circ 3}} = 1$ and $\|\frac{d\nabla_y g(x_1,y_1)}{d[y_1]^{\circ 3}} - \frac{d\nabla_y g(x_2,y_2)}{d[y_2]^{\circ 3}}\| = 0$. Therefore, $l_{g,2}$ can take value $0$ only for this assumption. To make $l_{g,2}$ consistent with other assumptions, we can have $l_{g,2} = 1$.

- Assumption 3.2 (vi): $\|\frac{d\nabla_y g(x,y)}{d[y]^{\circ 3}}\| = 1$, so $C = 1$.

- Assumption 3.3 (i): $\nabla_x f(x, y) = 0$, so $l_{f,1} = 0$.

- Assumption 3.3 (ii): $\frac{df(x,y)}{d[y]^{\circ 3}} = 1$, so $\|\frac{df(x_1,y_1)}{d[y_1]^{\circ 3}} - \frac{df(x_2,y_2)}{d[y_2]^{\circ 3}}\| = 0$, so $l_{f,1} = 0$.

- Assumption 3.3 (iii): $\|\frac{df(x,y)}{d[y]^{\circ 3}}\| = 1$, so $l_{f,0} = 1$.

- Assumption 3.3 (iv): $\nabla_y g(x, y^*(x)) = (y^*(x))^3 - \sin x = 0$, so $y^*(x) = (\sin x)^{\frac{1}{3}}$, therefore $\Phi(x) = \sin x$ and $\Delta_\Phi \leq 2$.

**Example 2.**    In the data hypercleaning task, choose $\mathcal{L}(w, \zeta)$ in Eq. (5) to be

$$
\mathcal{L}(w; \zeta_i) = \left| x_i^\top w - \bar{y}_i \right|^p, \qquad \zeta_i = (x_i, \bar{y}_i) \quad i \in [n]. \tag{12}
$$

Then the lower-level objective is

$$
g(w, \lambda) = \frac{1}{n} \left\| \Lambda \left( Xw - \bar{y} \right) \right\|_p^p + c \left\| w \right\|_p^p, \tag{13}
$$

where $w$ is the lower-level variable and $\lambda$ is the upper-level variable, and

$$\Lambda = \mathrm{diag}\big(\sigma(\lambda_1)^{1/p},\ldots,\sigma(\lambda_n)^{1/p}\big), \quad X = \begin{bmatrix} x_1^\top \\ \vdots \\ x_n^\top \end{bmatrix} \in \mathbb{R}^{n\times d}, \quad \bar{y} = \begin{bmatrix} \bar{y}_1 \\ \vdots \\ \bar{y}_n \end{bmatrix} \in \mathbb{R}^n, \quad w \in \mathbb{R}^d.$$

Write $g(\cdot,\lambda) = G(\cdot) + R(\cdot)$ with

$$G(w) := \frac{1}{n}\left\|\Lambda(Xw - \bar{y})\right\|_p^p, \qquad R(w) := c\|w\|_p^p.$$

By Assumption 3.2 (i), the sum of a $(\mu_1, p)$-uniformly-convex function and a $(\mu_2, p)$-uniformly-convex function is $(\mu_1 + \mu_2, p)$-uniformly-convex. We now identify $\mu_1$ and $\mu_2$.

By Eq. (16), we know that $c\|w\|_p^p$ is $\big(\frac{cp}{d^{1/2-1/p}}, p\big)$-uniformly convex. Hence $\mu_2 = \frac{cp}{d^{1/2-1/p}}$.

By translation invariance of uniform convexity, it suffices to consider $\frac{1}{n}\|\Lambda Xw\|_p^p$. Using the $p$-minimum singular value

$$\sigma_{\min,p}(M) := \inf_{\|u\|_p=1} \|Mu\|_p,$$

together with standard $\ell_p - \ell_2$ norm transitions for $p \geq 2$, we obtain the lower bound

$$\frac{1}{n}\left\|\Lambda Xw\right\|_p^p \geq \frac{\big(\sigma_{\min,p}(\Lambda X)\big)^p}{n\,d^{1/2-1/p}}\|w\|_2^p. \tag{14}$$

Therefore $G$ is $\big(\mu_1, p\big)$-uniformly convex with $\mu_1 = \frac{p\big(\sigma_{\min,p}(\Lambda X)\big)^p}{n\,d^{1/2-1/p}}$.

Combining the two parts via assumption 3.2 (i), the function $g$ in Eq. (13) is $(\mu, p)$-uniformly convex with

$$\mu = \frac{p\big(\sigma_{\min,p}(\Lambda X)\big)^p}{n\,d^{1/2-1/p}} + \frac{cp}{d^{1/2-1/p}}.$$

This establishes LLUC for the hypercleaning lower-level objective and quantifies its modulus.

**Example 3.** Let $p \geq 2$ be an even integer, and let the functions $f$ and $g$ be defined as:

$$f(x,y) = \begin{cases} -1 & y < -\left(\frac{\pi}{2}\right)^{\frac{1}{p-1}} \\ \sin(y^{p-1}) & y \in \left[-\left(\frac{\pi}{2}\right)^{\frac{1}{p-1}}, \left(\frac{\pi}{2}\right)^{\frac{1}{p-1}}\right] , \\ 1 & y > \left(\frac{\pi}{2}\right)^{\frac{1}{p-1}} \end{cases} \quad g(x,y) = \frac{1}{p}y^p - y\sin x. \tag{15}$$

Now we verify the assumptions.

- Assumption 3.2 (i): Note that $\frac{1}{p}y^p$ is a $(1,p)$ uniform convex function, $y\sin x$ is a linear function with $y$, so $g(x,y) = \frac{1}{p}y^p - y\sin x$ is a $(1,p)$ uniform convex with $y$.

- Assumption 3.2 (ii): $\|\nabla_{yy}g(x,y)\| = (p-1)y^{p-2} \leq 4(p-1) + 2(p-1)\|y^{p-1} - \sin x\| = 4(p-1) + 2(p-1)\|\nabla_y g(x,y)\|$, hence we have $L_0 = 4(p-1)$, $L_1 = 2(p-1)$.

- Assumption 3.2 (iii): $\nabla_y g(x,y) = y^{p-1} - \sin x$, so $\|\nabla_y g(x_1,y) - \nabla_y g(x_2,y)\| = \|\sin x_2 - \sin x_1\| \leq \|x_1 - x_2\|$. Therefore $l_{g,1} = 1$.

- Assumption 3.2 (iv): $\nabla_{xy}g(x,y) = -\cos x$, so $\|\nabla_{xy}g(x_1,y_1) - \nabla_{xy}g(x_2,y_2)\| = \|\cos x_2 - \cos x_1\| \leq \|x_1 - x_2\|$. Therefore $l_{g,2} = 1$.

- Assumption 3.2 (v): $\nabla_y g(x,y) = y^{p-1} - \sin x$, so $\frac{d\nabla_y g(x,y)}{d[y]^{\circ p-1}} = 1$ and $\|\frac{d\nabla_y g(x_1,y_1)}{d[y_1]^{\circ p-1}} - \frac{d\nabla_y g(x_2,y_2)}{d[y_2]^{\circ p-1}}\| = 0$. Therefore, $l_{g,2}$ can take value $0$ only for this assumption. To make $l_{g,2}$ consistent with other assumptions, we can have $l_{g,2} = 1$.

- Assumption 3.2 (vi): $\|\frac{d\nabla_y g(x,y)}{d[y]^{\circ p-1}}\| = 1$, so $C = 1$.

- Assumption 3.3 (i): $\nabla_x f(x,y) = 0$, so $l_{f,1} = 0$. To make $l_{f,1}$ consistent with other assumptions, we can have $l_{f,1} = (p-1)\left(\frac{\pi}{2}\right)^{\frac{p-2}{p-1}}$.

- Assumption 3.3 (ii): $\frac{df(x,y)}{d[y]^{\circ p-1}} = \begin{cases} 0, & y > (\frac{\pi}{2})^{\frac{1}{p-1}} \\ \cos(y^{p-1}), & -(\frac{\pi}{2})^{\frac{1}{p-1}} \le y \le (\frac{\pi}{2})^{\frac{1}{p-1}} \\ 0, & y < -(\frac{\pi}{2})^{\frac{1}{p-1}} \end{cases}$

  so from the mean-value theorem, we have

  $$\left\| \frac{df(x_1,y_1)}{d[y_1]^{\circ p-1}} - \frac{df(x_2,y_2)}{d[y_2]^{\circ p-1}} \right\| \le \max_{y \in [-(\frac{\pi}{2})^{\frac{1}{p-1}}, (\frac{\pi}{2})^{\frac{1}{p-1}}]} (p-1)y^{p-2} \sin(y^{p-1}) \|y_1 - y_2\| \le (p-1)(\frac{\pi}{2})^{\frac{p-2}{p-1}} \|y_1 - y_2\|,$$

  and hence $l_{f,1} = (p-1)\left(\frac{\pi}{2}\right)^{\frac{p-2}{p-1}}$.

- Assumption 3.3 (iii): $\left\| \frac{df(x,y)}{d[y]^{\circ p-1}} \right\| \le 1$, so $l_{f,0} = 1$.

- Assumption 3.3 (iv): $\nabla_y g(x, y^*(x)) = (y^*(x))^{p-1} - \sin x = 0$, so $y^*(x) = (\sin x)^{\frac{1}{p-1}}$, therefore $\Phi(x) = \sin \sin x$ and $\Delta_\phi = 2$.

**Example 4.** Define $x = (x_1, \ldots, x_d) \in \mathbb{R}^d$, $y = (y_1, \ldots, y_d) \in \mathbb{R}^d$, $p$ is an even integer or a fraction of even number divide by an old number. Then we consider the following function

$$f(x, y) = \sum_{i=1}^d |y_i|^{p-1} sgn(y_i), \quad g(x, y) = \frac{1}{p} \|y\|_p^p - \sum_{i=1}^d y_i \sin x_i,$$

where $sgn(\cdot)$ is the sign function, $p \ge 2$ is even number.

Define $y^*(x) = (y_1^*(x), \ldots, y_d^*(x)) := (y_1^*, \ldots, y_d^*)$. Note that $\nabla_y g(x, y^*(x)) = 0$, therefore we have $(|y_1^*|^{p-1} sgn(y_1^*), \ldots, |y_d^*|^{p-1} sgn(y_d^*)) = (\sin x_1, \ldots, \sin x_d)$ and $\Phi(x) = \sum_{i=1}^d |y_i^*|^{p-1} sgn(y_i) = \sum_{i=1}^d \sin x_i$.

All assumptions can be satisfied by choosing the problem-dependent parameters as the following:

| $p$ | $\mu$ | $L_0$ | $L_1$ | $l_{g,1}$ | $l_{g,2}$ | $C$ | $l_{f,1}$ | $l_{f,0}$ | $\Delta_\phi$ |
|---|---|---|---|---|---|---|---|---|---|
| $p$ | $\frac{1}{d^{\frac{1}{2}-\frac{1}{p}}}$ | $4(p-1)$ | $2(p-1)$ | $1$ | $1$ | $1$ | $0$ | $\sqrt{d}$ | $2d$ |

Table 1: Parameter values as functions of $p$ and $d$

- Assumption 3.2 (i): $g(x, y)$ is $\left( \frac{1}{d^{\frac{1}{2}-\frac{1}{p}}}, p \right)$ uniform-convex due to:

  $$\frac{1}{p} \|y\|_2^p \ge \frac{1}{p} \|y\|_p^p \ge \frac{1}{p d^{\frac{1}{2}-\frac{1}{p}}} \|y\|_2^p. \tag{16}$$

- Assumption 3.2 (ii): $\nabla_{yy} g(x, y) = \text{diag}\left\{ (p-1)y_1^{p-2}, \ldots, (p-1)y_d^{p-2} \right\}$ and $g(x, y)$ is $(4(p-1), 2(p-1))$-smooth w.r.t y:

  $$\begin{aligned} \|\nabla_{yy} g(x, y)\|_2 &= (p-1)\|[y]^{\circ(p-2)}\|_\infty \\ &\le 4(p-1) + 2(p-1)\|[y]^{\circ(p-1)} - \sin(x))\|_\infty \\ &\le 4(p-1) + 2(p-1)\|\nabla_y g(x, y)\|_\infty \\ &\le 4(p-1) + 2(p-1)\|\nabla_y g(x, y)\|_2. \end{aligned}$$

- Assumption 3.2 (iii): The gradient $\nabla_y g(x, y) = [y]^{\circ(p-1)} - \sin(x)$ is 1-Lipschitz continuous w.r.t. $x$.

- Assumption 3.2 (iv) $\nabla_{xy} g(x, y) = -\cos(x))$ is 1-jointly Lipschitz w.r.t. $(x, y)$.

- Assumption 3.2 (v) and (vi): $\frac{d\nabla_y g(x,y)}{d[y]^{\circ(p-1)}} = I$ is 0-jointly Lipschitz w.r.t. $(x, y)$, and it satisfies the uniform bound:

  $$\left\| \frac{d\nabla_y g(x, y)}{d[y]^{\circ(p-1)}} \right\|_2 = \lambda_{\max}(I) = 1.$$

- Assumption 3.3 (i): $\nabla_x f(x, y) = 0$ jointly Lipschitz w.r.t. $(x, y)$.

- Assumption 3.3 (ii) and (iii): $\frac{df(x,y)}{d[y]^{\circ(p-1)}} = \mathbf{1}$ is 0-jointly Lipschitz and satisfies the uniform bound:

$$\left\| \frac{df(x, y)}{d[y]^{\circ(p-1)}} \right\|_2 \leq \sqrt{d}$$

- Assumption 3.3 (iv): $\Phi(x_0) - \inf \Phi \leq 2d = \Delta_\phi$.

# B    PROOFS IN SECTION 4

## B.1    PROOF OF LEMMA 4.2

**Lemma B.1** (Restatement of Lemma 4.2). *$y^*(x)$ is hölder continuous: for any $x_1, x_2 \in \mathbb{R}^{d_x}$, we have*

$$\|y^*(x_2) - y^*(x_1)\| \leq l_p \|x_2 - x_1\|^{\frac{1}{p-1}}, \qquad where \quad l_p = \left( \frac{p l_{g,1}}{\mu} \right)^{\frac{1}{p-1}}. \tag{17}$$

*Proof of Theorem B.1.* Since $g(x, \cdot)$ is uniformly convex, for any $y \in \mathbb{R}^{d_y}$ we have the following $p$-th order growth condition:

$$g(x_1, y) \geq g(x_1, y^*(x_1)) + \langle \nabla_y g(x_1, y^*(x_1)), y - y_1 \rangle + \frac{\mu}{p} \|y - y_1\|^p$$
$$= g(x_1, y^*(x_1)) + \frac{\mu}{p} \|y - y^*(x_1)\|^p. \tag{18}$$

In particular, if we let $y = y^*(x_2)$, then

$$g(x_1, y^*(x_2)) - g(x_1, y^*(x_1)) \geq \frac{\mu}{p} \|y^*(x_2) - y^*(x_1)\|^p. \tag{19}$$

Next, we follow the similar procedure as in proof of Proposition 4.32 in Bonnans & Shapiro (2013). We consider the difference function $h(y) := g(x_2, y) - g(x_1, y)$, then we have

$$g(x_1, y^*(x_2)) - g(x_1, y^*(x_1)) = h(y^*(x_1)) - h(y^*(x_2)) + g(x_2, y^*(x_2)) - g(x_2, y^*(x_1))$$
$$\leq h(y^*(x_1)) - h(y^*(x_2)) \leq l_{g,1} \|x_2 - x_1\| \cdot \|y^*(x_2) - y^*(x_1)\| \tag{20}$$

where in the first inequality we use $g(x_2, y^*(x_2)) \leq g(x_2, y^*(x_1))$, and in the second inequality we use the fact that $g$ is $l_{g,1}$-smooth in $x$ and mean value theorem to obtain (denote $\kappa(x_1, x_2)$ as the Lipschitz constant of function $h$):

$$\kappa(x_1, x_2) \leq \sup_{y \in \mathbb{R}^{d_y}} \|\nabla h(y)\| = \sup_{y \in \mathbb{R}^{d_y}} \|\nabla_y g(x_1, y) - \nabla_y g(x_2, y)\| \leq l_{g,1} \|x_1 - x_2\| \tag{21}$$

Combining Eq. (19) and Eq. (20) yields

$$\frac{\mu}{p} \|y^*(x_2) - y^*(x_1)\|^p \leq l_{g,1} \|x_2 - x_1\| \cdot \|y^*(x_2) - y^*(x_1)\|.$$

Therefore, the Lemma is proved. $\qquad\qquad\qquad\qquad\qquad\qquad\qquad\qquad\qquad\qquad\square$

## B.2    A TECHNICAL LEMMA UNDER A DIFFERENT ASSUMPTION

**Lemma B.2** (Positive Definite Generalized Hessian). *$\frac{d\nabla_y g(x,y)}{d[y]^{\circ p-1}}$ is an invertible matrix and $\lambda_{\min}\left( \frac{d\nabla_y g(x,y)}{d[y]^{\circ p-1}} \right) \geq \mu$, where $\lambda_{\min}(\cdot)$ denotes the minimum eigenvalue of a matrix.*

**Remark:** If we do not directly assume the generalized Hessian is positive definite, under the assumption that $\frac{d\nabla_y g(x,y)}{d[y]^{\circ p-1}}$ is independent of $y^{\circ(p-1)}$, Lemma B.2 provides a characterization of the minimum eigenvalue of a generalized Hessian matrix, which plays a crucial role in establishing our implicit function theorem under the LLUC condition.

*Proof.* Define $z = [y]^{\circ p-1}$. Since $\frac{d\nabla_y g(x,y)}{d[y]^{\circ p-1}}$ exists, then by Definition A.1, we have for any $\bar{h} \in \mathbb{R}^{d_y}$ and any $z \in \mathbb{R}^{d_y}$, there exists a linear map $J_2 := \frac{d\nabla_y g(x,y)}{d[y]^{\circ p-1}} \in \mathbb{R}^{d_y \times d_y}$ such that the following holds

$$\lim_{\delta \to 0} \frac{\nabla_y g(x, [z+\delta\bar{h}]^{\circ \frac{1}{p-1}}) - \nabla_y g(x, z^{\circ \frac{1}{p-1}}) - \langle J_2, \delta\bar{h} \rangle}{\|\delta\bar{h}\|} = 0. \tag{22}$$

Since $J_2$ is independent of $z$ (by definition A.1), we can take $z = 0$ in Eq. (22), rearrange this equality and take norm on both sides, we have

$$\lim_{\delta \to 0} \frac{\|\nabla_y g(x, [\delta\bar{h}]^{\circ \frac{1}{p-1}}) - \nabla_y g(x, 0)\|}{\|\delta\bar{h}\|} = \lim_{\delta \to 0} \frac{\|J_2 \delta\bar{h}\|}{\|\delta\bar{h}\|}. \tag{23}$$

By uniform convexity of $g$ in terms of $y$, we have

$$\|\nabla_y g(x, [\delta\bar{h}]^{\circ \frac{1}{p-1}}) - \nabla_y g(x, 0)\| \geq \mu \|[\delta\bar{h}]^{\circ \frac{1}{p-1}}\|^{p-1} \geq \mu \|[\delta\bar{h}]^{\circ \frac{1}{p-1}}\|_{2(p-1)}^{p-1} = \mu \|\delta\bar{h}\|. \tag{24}$$

where the first inequality holds because of the uniform convexity, the second inequality holds by the fact that $\|y\| \geq \|y\|_{2(p-1)}$ for $p \geq 2$, and the last equality holds by the definition of $2(p-1)$-norm.

Combining Eq. (23) and Eq. (24), we have

$$\lim_{\delta \to 0} \frac{\|J_2 \delta\bar{h}\|}{\|\delta\bar{h}\|} \geq \mu. \tag{25}$$

Since $\bar{h}$ can be a vector with any direction, therefore $J_2 = \frac{d\nabla_y g(x,y)}{d[y]^{\circ p-1}}$ is an invertible matrix and $\lambda_{\min}\left(\frac{d\nabla_y g(x,y)}{d[y]^{\circ p-1}}\right) \geq \mu$.

$\square$

### B.3 PROOF OF THEOREM 4.1

**Theorem B.3** (Restatement of Theorem 4.1). *Suppose Assumption 3.2 and 3.3 hold. Then $\Phi$ is differentiable in $x$ and can be computed as the following:*

$$\nabla\Phi(x) = \nabla_x f(x, y^*(x)) - \nabla_{xy} g(x, y^*(x)) \left[\frac{d\nabla_y g(x, y^*(x))}{d[y^*(x)]^{\circ p-1}}\right]^{-1} \frac{df(x, y^*(x))}{d[y^*(x)]^{\circ p-1}}. \tag{26}$$

*In addition, the function $\Phi$ satisfies the following properties:*

$$\|\nabla\Phi(x_1) - \nabla\Phi(x_2)\| \leq L_{\phi_1}\|x_1 - x_2\|^{\frac{1}{p-1}} + L_{\phi_2}\|x_1 - x_2\|, \tag{27}$$

$$\Phi(x_1) \leq \Phi(x_2) + \langle \nabla\Phi(x_2), x_1 - x_2 \rangle + \frac{(p-1)L_{\phi_1}}{p}\|x_1 - x_2\|^{\frac{p}{p-1}} + \frac{L_{\phi_2}}{2}\|x_1 - x_2\|^2. \tag{28}$$

*where* $l_p = \left(\frac{p l_{g,1}}{\mu}\right)^{\frac{1}{p-1}}$, $L_{\phi_1} = l_p(l_{f,1} + \frac{l_{f,2} l_{g,2}}{\mu} + \frac{l_{g,1} l_{f,1}}{\mu} + \frac{l_{g,1} l_{f,1} l_{g,2}}{\mu^2})$, $L_{\phi_2} = l_{f,1} + \frac{l_{f,2} l_{g,2}}{\mu} + \frac{l_{g,1} l_{f,1}}{\mu} + \frac{l_{g,1} l_{f,1} l_{g,2}}{\mu^2}$.

*Proof.* Define $y^*(x) = [z^*(x)]^{\circ \frac{1}{p-1}}$. Noting that $\nabla_y g(x, y^*(x)) = 0$, we take derivative in terms of $x$ on both sides and use the chain rule, which yields

$$\nabla_{xy} g(x, [z^*(x)]^{\circ \frac{1}{p-1}}) + \frac{dz^*(x)}{dx} \frac{d\nabla_y g(x, [z^*(x)]^{\circ \frac{1}{p-1}})}{dz^*(x)} = 0. \tag{29}$$

Therefore,

$$\nabla_{xy} g(x, y^*(x)) + \frac{dz^*(x)}{dx} \frac{d\nabla_y g(x, y^*(x))}{dz^*(x)} = 0. \tag{30}$$

Now we start to derive the properties of $\Phi$.

By Lemma B.2, we know that $\lambda_{\min}(\frac{d\nabla_y g(x,y)}{d[y]^{\circ p-1}}) \geq \mu > 0$ holds for any $y$, therefore we plug in $y = y^*(x)$ and know that $\frac{d\nabla_y g(x,y^*(x))}{dz^*(x)}$ is a invertible matrix. Hence we have

$$\frac{dz^*(x)}{dx} = -\nabla_{xy}g(x,y^*(x))\left[\frac{d\nabla_y g(x,y^*(x))}{dz^*(x)}\right]^{-1}. \tag{31}$$

Therefore, $z^*(x)$ is differentiable with $x$ everywhere.

By Assumption 3.3 (iii), we know that $J_1 = \frac{df(x,[z^*(x)]^{\circ\frac{1}{p-1}})}{dz^*(x)}$ exists. Therefore, we can use chain rule to directly derive hypergradient formula:

$$\nabla\Phi(x) = \frac{df(x,y^*(x))}{dx} = \nabla_x f(x,[z^*(x)]^{\circ\frac{1}{p-1}}) + \frac{dz^*(x)}{dx}\frac{df(x,[z^*(x)]^{\circ\frac{1}{p-1}})}{dz^*(x)}$$

$$= \nabla_x f(x,y^*(x)) - \nabla_{xy}g(x,y^*(x))\left[\frac{d\nabla_y g(x,y^*(x))}{dz^*(x)}\right]^{-1}\frac{df(x,y^*(x))}{dz^*(x)}$$

$$= \nabla_x f(x,y^*(x)) - \nabla_{xy}g(x,y^*(x))\left[\frac{d\nabla_y g(x,y^*(x))}{d[y^*(x)]^{\circ p-1}}\right]^{-1}\frac{df(x,y^*(x))}{d[y^*(x)]^{\circ p-1}}. \tag{32}$$

Therefore, the final hypergradient can be computed as:

$$\nabla\Phi(x) = \nabla_x f(x,y^*(x)) - \nabla_{xy}g(x,y^*(x))\left[\frac{d\nabla_y g(x,y^*(x))}{d[y^*(x)]^{\circ p-1}}\right]^{-1}\frac{df(x,y^*(x))}{d[y^*(x)]^{\circ p-1}}. \tag{33}$$

Define

$$v(x,y) := -\nabla_{xy}g(x,y)\left[\frac{d\nabla_y g(x,y)}{d[y]^{\circ p-1}}\right]^{-1}\frac{df(x,y)}{d[y]^{\circ p-1}}. \tag{34}$$

Now we start to prove the properties of $\Phi$. By Assumption 3.2 (iii), we have for any $x_1, x_2 \mathbb{R}^{d_x}$, the following inequality holds:

$$\|\nabla_y g(x_1,y) - \nabla_y g(x_2,y)\| \leq l_{g,1}\|x_1 - x_2\| \implies \|\nabla_{xy}g(x,y)\| \leq l_{g,1}. \tag{35}$$

so we have

$$\left\|\frac{dz^*(x)}{dx}\right\| = \left\|\frac{d[y^*(x)]^{\circ p-1}}{dx}\right\| \leq \|\nabla_{xy}g(x,y^*(x))\|\left\|\left[\frac{d\nabla_y g(x,y^*(x))}{dz^*(x)}\right]^{-1}\right\| \leq \frac{l_{g,1}}{\mu}. \tag{36}$$

In addition, note that for any invertible matrices $H_1$ and $H_2$, the inequality holds:

$$\|H_2^{-1} - H_1^{-1}\| = \|H_1^{-1}(H_1 - H_2)H_2^{-1}\| \leq \|H_1^{-1}\|\|H_2^{-1}\|\|H_1 - H_2\|, \tag{37}$$

therefore we have

$$\left\|\left[\frac{d\nabla_y g(x_1,y^*(x_1))}{d[y^*(x_1)]^{\circ p-1}}\right]^{-1} - \left[\frac{d\nabla_y g(x_2,y^*(x_2))}{d[y^*(x_2)]^{\circ p-1}}\right]^{-1}\right\| \leq \frac{1}{\mu^2}\left\|\frac{d\nabla_y g(x_1,y^*(x_1))}{d[y^*(x_1)]^{\circ p-1}} - \frac{d\nabla_y g(x_2,y^*(x_2))}{d[y^*(x_2)]^{\circ p-1}}\right\|$$

$$\leq \frac{l_{g,2}}{\mu^2}\left(\|x_1 - x_2\| + \|y^*(x_1) - y^*(x_2)\|\right), \tag{38}$$

where the last inequality holds because of the $l_{g,2}$-jointly Lipschitz in $(x,y)$ for the matrix $\frac{d\nabla_y g(x,y)}{d[y]^{\circ p-1}}$ (i.e., Assumption 3.2 (v)).

For the second part of hypergradient, we have

$$\|v(x_1,y^*(x_1)) - v(x_2,y^*(x_2))\|$$

$$= \left\|\nabla_{xy}g(x_2,y^*(x_2))\left[\frac{d\nabla_y g(x_2,y^*(x_2))}{d[y^*(x_2)]^{\circ p-1}}\right]^{-1}\frac{df(x_2,y^*(x_2))}{d[y^*(x_2)]^{\circ p-1}} - \nabla_{xy}g(x_1,y^*(x_1))\left[\frac{d\nabla_y g(x_1,y^*(x_1))}{d[y^*(x_1)]^{\circ p-1}}\right]^{-1}\frac{df(x_1,y^*(x_1))}{d[y^*(x_1)]^{\circ p-1}}\right\|$$

$$= \left\|\nabla_{xy}g(x_2,y^*(x_2))\left[\frac{d\nabla_y g(x_2,y^*(x_2))}{d[y^*(x_2)]^{\circ p-1}}\right]^{-1}\frac{df(x_2,y^*(x_2))}{d[y^*(x_2)]^{\circ p-1}} - \nabla_{xy}g(x_1,y^*(x_1))\left[\frac{d\nabla_y g(x_2,y^*(x_2))}{d[y^*(x_2)]^{\circ p-1}}\right]^{-1}\frac{df(x_2,y^*(x_2))}{d[y^*(x_2)]^{\circ p-1}}\right.$$

$$+ \nabla_{xy}g(x_1, y^*(x_1)) \left[ \frac{d\nabla_y g(x_2, y^*(x_2))}{d[y^*(x_2)]^{\circ p-1}} \right]^{-1} \frac{df(x_2, y^*(x_2))}{d[y^*(x_2)]^{\circ p-1}} - \nabla_{xy}g(x_1, y^*(x_1)) \left[ \frac{d\nabla_y g(x_1, y^*(x_1))}{d[y^*(x_1)]^{\circ p-1}} \right]^{-1} \frac{df(x_1, y^*(x_1))}{d[y^*(x_1)]^{\circ p-1}} \Big\|$$

$$\leq \left\| \nabla_{xy}g(x_2, y^*(x_2)) \left[ \frac{d\nabla_y g(x_2, y^*(x_2))}{d[y^*(x_2)]^{\circ p-1}} \right]^{-1} \frac{df(x_2, y^*(x_2))}{d[y^*(x_2)]^{\circ p-1}} - \nabla_{xy}g(x_1, y^*(x_1)) \left[ \frac{d\nabla_y g(x_2, y^*(x_2))}{d[y^*(x_2)]^{\circ p-1}} \right]^{-1} \frac{df(x_2, y^*(x_2))}{d[y^*(x_2)]^{\circ p-1}} \right\|$$

$$+ \left\| \nabla_{xy}g(x_1, y^*(x_1)) \left[ \frac{d\nabla_y g(x_2, y^*(x_2))}{d[y^*(x_2)]^{\circ p-1}} \right]^{-1} \frac{df(x_2, y^*(x_2))}{d[y^*(x_2)]^{\circ p-1}} - \nabla_{xy}g(x_1, y^*(x_1)) \left[ \frac{d\nabla_y g(x_1, y^*(x_1))}{d[y^*(x_1)]^{\circ p-1}} \right]^{-1} \frac{df(x_1, y^*(x_1))}{d[y^*(x_1)]^{\circ p-1}} \right\|$$

$$\overset{(a)}{\leq} \frac{l_{f,0}}{\mu} \| \nabla_{xy}g(x_2, y^*(x_2)) - \nabla_{xy}g(x_1, y^*(x_1)) \|$$

$$+ l_{g,1} \left\| \left[ \frac{d\nabla_y g(x_1, y^*(x_1))}{d[y^*(x_1)]^{p-1}} \right]^{-1} \frac{df(x_1, y^*(x_1))}{d[y^*(x_1)]^{\circ p-1}} - \left[ \frac{d\nabla_y g(x_2, y^*(x_2))}{d[y^*(x_2)]^{p-1}} \right]^{-1} \frac{df(x_2, y^*(x_2))}{d[y^*(x_2)]^{\circ p-1}} \right\|$$

$$\overset{(b)}{\leq} \frac{l_{f,0}}{\mu} \| \nabla_{xy}g(x_2, y^*(x_2)) - \nabla_{xy}g(x_1, y^*(x_1)) \| + l_{g,1} \left\| \frac{df(x_1, y^*(x_1))}{d[y^*(x_1)]^{\circ p-1}} \right\| \left\| \left[ \frac{d\nabla_y g(x_1, y^*(x_1))}{d[y^*(x_1)]^{\circ p-1}} \right]^{-1} - \left[ \frac{d\nabla_y g(x_2, y^*(x_2))}{d[y^*(x_2)]^{\circ p-1}} \right]^{-1} \right\|$$

$$+ l_{g,1} \left\| \left[ \frac{d\nabla_y g(x_2, y^*(x_2))}{d[y^*(x_2)]^{\circ p-1}} \right]^{-1} \right\| \left\| \frac{df(x_1, y^*(x_1))}{d[y^*(x_1)]^{\circ p-1}} - \frac{df(x_2, y^*(x_2))}{d[y^*(x_2)]^{\circ p-1}} \right\|$$

$$\overset{(c)}{\leq} \frac{l_{f,0}}{\mu} \| \nabla_{xy}g(x_2, y^*(x_2)) - \nabla_{xy}g(x_1, y^*(x_1)) \| + l_{g,1}l_{f,0} \left\| \left[ \frac{d\nabla_y g(x_1, y^*(x_1))}{d[y^*(x_1)]^{\circ p-1}} \right]^{-1} - \left[ \frac{d\nabla_y g(x_2, y^*(x_2))}{d[y^*(x_2)]^{p-1}} \right]^{-1} \right\|$$

$$+ \frac{l_{g,1}}{\mu} \left\| \frac{df(x_1, y^*(x_1))}{d[y^*(x_1)]^{\circ p-1}} - \frac{df(x_2, y^*(x_2))}{d[y^*(x_2)]^{\circ p-1}} \right\|$$

$$\overset{(d)}{\leq} \left( \frac{l_{f,0}l_{g,2}}{\mu} + \frac{l_{g,1}l_{f,1}}{\mu} + \frac{l_{g,1}l_{f,0}l_{g,2}}{\mu^2} \right) (\|x_1 - x_2\| + \|y^*(x_1) - y^*(x_2)\|), \tag{39}$$

where (a) holds because of Assumption 3.3 (iii), Lemma B.2 and Eq. (35); (b) holds because of triangle inequality of the norm, (c) holds because of Assumption 3.3 (iii) and Lemma B.2; (d) holds because of Assumption 3.2 (iv), Assumption 3.3 (ii) and Eq. (38).

Therefore, the hypergradient satisfies the following property:

$$\|\nabla\Phi(x_1) - \nabla\Phi(x_2)\| = \|\nabla_x f(x_1, y^*(x_1)) + v(x_1, y^*(x_1)) - [\nabla_x f(x_2, y^*(x_2)) + v(x_2, y^*(x_2))]\|$$

$$\leq l_{f,1}(\|x_1 - x_2\| + \|y^*(x_1) - y^*(x_2)\|) + \|v(x_1, y^*(x_1) - v(x_2, y^*(x_2))\|$$

$$\leq (l_{f,1} + \frac{l_{f,0}l_{g,2}}{\mu} + \frac{l_{g,1}l_{f,1}}{\mu} + \frac{l_{g,1}l_{f,0}l_{g,2}}{\mu^2})\|x_1 - x_2\| + (l_{f,1} + \frac{l_{f,0}l_{g,2}}{\mu} + \frac{l_{g,1}l_{f,1}}{\mu} + \frac{l_{g,1}l_{f,0}l_{g,2}}{\mu^2}))\|y^*(x_1) - y^*(x_2)\|$$

$$\leq (l_{f,1} + \frac{l_{f,0}l_{g,2}}{\mu} + \frac{l_{g,1}l_{f,1}}{\mu} + \frac{l_{g,1}l_{f,0}l_{g,2}}{\mu^2})\|x_1 - x_2\| + (l_{f,1} + \frac{l_{f,0}l_{g,2}}{\mu} + \frac{l_{g,1}l_{f,1}}{\mu} + \frac{l_{g,1}l_{f,0}l_{g,2}}{\mu^2}))l_p\|x_1 - x_2\|^{\frac{1}{p-1}} \tag{40}$$

Define $L_{\phi_1} := l_p(l_{f,1} + \frac{l_{f,0}l_{g,2}}{\mu} + \frac{l_{g,1}l_{f,1}}{\mu} + \frac{l_{g,1}l_{f,0}l_{g,2}}{\mu^2})$ and $L_{\phi_2} := l_{f,1} + \frac{l_{f,0}l_{g,2}}{\mu} + \frac{l_{g,1}l_{f,1}}{\mu} + \frac{l_{g,1}l_{f,0}l_{g,2}}{\mu^2}$. Then we have

$$\|\nabla\Phi(x_1) - \nabla\Phi(x_2)\| \leq L_{\phi_1}\|x_1 - x_2\|^{\frac{1}{p-1}} + L_{\phi_2}\|x_1 - x_2\|. \tag{41}$$

Furthermore, we have

$$\Phi(x_1) - \Phi(x_2) - \langle \nabla\Phi(x_2), x_1 - x_2 \rangle = \int_0^1 \langle \nabla\Phi(x_2 + t(x_1 - x_2)) - \nabla\Phi(x_2), x_1 - x_2 \rangle dt$$

$$\leq \int_0^1 \|\nabla\Phi(x_2 + t(x_1 - x_2)) - \nabla\Phi(x_2)\| \|x_1 - x_2\| dt$$

$$\leq \|x_1 - x_2\|^{\frac{p}{p-1}} \int_0^1 (L_{\phi_1}t^{\frac{1}{p-1}})dt + \|x_1 - x_2\|^2 \int_0^1 (L_{\phi_2}t)dt$$

$$= \frac{(p-1)L_{\phi_1}}{p}\|x_1 - x_2\|^{\frac{p}{p-1}} + \frac{L_{\phi_2}}{2}\|x_1 - x_2\|^2. \tag{42}$$

$$\square$$

### B.4 GENERALIZATION OF ASSUMPTIONS

If there exists a constant $a$ such that $\frac{df(x,y)}{d[y-a]^{\circ p-1}}$, $\frac{d\nabla_y g(x,y)}{d[y-a]^{\circ p-1}}$ exist and satisfy all of our assumptions, we can choose $z = [y - a]^{\circ p-1}$, then $y^*(x) = [z^*(x)]^{\circ \frac{1}{p-1}} + a$ and we can derive the same hypergradient formula. Therefore we assume $a = 0$ without loss of generality. To show the fact that the hypergradient formula is the same as in the case of $a = 0$, we have

$$\nabla\Phi(x) = \frac{df(x, y^*(x))}{dx} = \nabla_x f(x, [z^*(x)]^{\circ \frac{1}{p-1}} + a) + \frac{dz^*(x)}{dx}\frac{df(x, [z^*(x)]^{\circ \frac{1}{p-1}} + a)}{dz^*(x)}$$

$$= \nabla_x f(x, [z^*(x)]^{\circ \frac{1}{p-1}} + a) - \nabla_{xy}g(x, [z^*(x)]^{\circ \frac{1}{p-1}} + a)\left[\frac{d(\nabla_y g(x, [z^*(x)]^{\circ \frac{1}{p-1}} + a)}{dz^*(x)}\right]^{-1}\frac{df(x, [z^*(x)]^{\circ \frac{1}{p-1}} + a)}{dz^*(x)}$$

$$= \nabla_x f(x, y^*(x)) - \nabla_{xy}g(x, y^*(x))\left[\frac{d\nabla_y g(x, y^*(x))}{d[y^*(x)]^{\circ p-1}}\right]^{-1}\frac{df(x, y^*(x))}{d[y^*(x)]^{\circ p-1}}.$$

### B.5 HYPERGRADIENT BIAS

**Lemma B.4** (Hypergradient Bias). *Suppose we have an inexact estimate $\hat{y}(x)$ for the optimal lower-level variable $y^*(x)$. Define $\widehat{\nabla}\Phi(x) = \nabla_x f(x, \hat{y}(x)) - \nabla_{xy}g(x, \hat{y}(x))\left[\frac{d\nabla_y g(x, \hat{y}(x))}{d[\hat{y}(x)]^{\circ p-1}}\right]^{-1}\frac{df(x, \hat{y}(x))}{d[\hat{y}(x)]^{\circ p-1}}$. Then we have*

$$\|\widehat{\nabla}\Phi(x) - \nabla\Phi(x)\| \le L_{\phi_2}\|\hat{y}(x) - y^*(x)\| \tag{43}$$

*where $L_{\phi_2} = l_{f,1} + \frac{l_{f,0}l_{g,2}}{\mu} + \frac{l_{g,1}l_{f,1}}{\mu} + \frac{l_{g,1}l_{f,0}l_{g,2}}{\mu^2}$.*

*Proof.* Similar to the proof of Theorem 4.1, we can use almost identical arguments to prove that $\nabla_x f(x, y) + v(x, y)$ is Lipschitz in $(x, y)$, where $v(x, y)$ is defined in Eq. (34). In particular, for any $x_1, x_2, y_1, y_2$, we can follow the similar analysis of Eq. (39) and leverage the $l_{f,1}$-joint Lipschitzness of $\nabla_x f(x, y)$ (i.e., Assumption 3.3 (i)) to show the following inequality holds:

$$\|\nabla_x f(x_1, y_1) + v(x_1, y_1) - \nabla_x f(x_2, y_2) - v(x_2, y_2)\|$$
$$\le \|\nabla_x f(x_1, y_1) - \nabla_x f(x_2, y_2)\| + \|v(x_1, y_1) - v(x_2, y_2)\|$$
$$\le l_{f,1}(\|x_1 - x_2\| + \|y_1 - y_2\|)$$
$$+ \left\|\nabla_{xy}g(x_2, y_2))\left[\frac{d\nabla_y g(x_2, y_2)}{d[y_2]^{\circ p-1}}\right]^{-1}\frac{df(x_2, y_2)}{d[y_2]^{\circ p-1}} - \nabla_{xy}g(x_1, y_1)\left[\frac{d\nabla_y g(x_1, y_1)}{d[y_1]^{\circ p-1}}\right]^{-1}\frac{df(x_1, y_1)}{d[y_1]^{\circ p-1}}\right\|$$
$$\le l_{f,1}(\|x_1 - x_2\| + \|y_1 - y_2\|) + \left\|\nabla_{xy}g(x_2, y_2))\left[\frac{d\nabla_y g(x_2, y_2)}{d[y_2]^{\circ p-1}}\right]^{-1}\frac{df(x_2, y_2)}{d[y_2]^{\circ p-1}} - \nabla_{xy}g(x_1, y_1)\left[\frac{d\nabla_y g(x_2, y_2)}{d[y_2]^{\circ p-1}}\right]^{-1}\frac{df(x_2, y_2)}{d[y_2]^{\circ p-1}}\right\|$$
$$+ \left\|\nabla_{xy}g(x_1, y_1)\left[\frac{d\nabla_y g(x_2, y_2)}{d[y_2]^{\circ p-1}}\right]^{-1}\frac{df(x_2, y_2)}{d[y_2]^{\circ p-1}} - \nabla_{xy}g(x_1, y_1)\left[\frac{d\nabla_y g(x_1, y_1)}{d[y_1]^{\circ p-1}}\right]^{-1}\frac{df(x_1, y_1)}{d[y_1]^{\circ p-1}}\right\|$$
$$\le l_{f,1}(\|x_1 - x_2\| + \|y_1 - y_2\|) + \frac{l_{g,2}}{\mu}(\|x_1 - x_2\| + \|y_1 - y_2\|)$$
$$+ l_{g,1}l_{f,0}\left\|\left[\frac{d\nabla_y g(x_2, y_2)}{d[y_2]^{\circ p-1}}\right]^{-1} - \left[\frac{d\nabla_y g(x_2, y_2)}{d[y_2]^{\circ p-1}}\right]^{-1}\right\| + \frac{l_{g,1}}{\mu}\left\|\frac{df(x_1, y_1)}{d[y_1]^{\circ p-1}} - \frac{df(x_1, y_1)}{d[y_1]^{\circ p-1}}\right\|$$
$$\le \left(\frac{l_{f,0}l_{g,2}}{\mu} + \frac{l_{g,1}l_{f,1}}{\mu} + \frac{l_{g,1}l_{f,0}l_{g,2}}{\mu^2}\right)(\|x_1 - x_2\| + \|y_1 - y_2\|) + l_{f,1}(\|x_1 - x_2\| + \|y_1 - y_2\|)$$
$$= L_{\phi_2}(\|x_1 - x_2\| + \|y_1 - y_2\|). \tag{44}$$

Therefore, we have

$$\|\widehat{\nabla}\Phi(x) - \nabla\Phi(x)\| \le \|\nabla_x f(x, \hat{y}(x)) - \nabla_x f(x, y^*(x))\| + \|v(x, \hat{y}(x)) - v(x, y^*(x))\|$$
$$\le l_{f,1}\|\hat{y}(x) - y^*(x)\| + \left(\frac{l_{f,0}l_{g,2}}{\mu} + \frac{l_{g,1}l_{f,1}}{\mu} + \frac{l_{g,1}l_{f,0}l_{g,2}}{\mu^2}\right)\|\hat{y}(x) - y^*(x)\| \tag{45}$$
$$= L_{\phi_2}\|\hat{y}(x) - y^*(x)\|.$$

Therefore the proof is done. □

## B.6 HYPERGRADIENT IMPLEMENTATION

**Lemma B.5.** *Denote $H$ as*

$$H := \frac{1}{C} \sum_{q=0}^{Q-1} \prod_{j=1}^{q} \left( I - \frac{1}{C} \frac{d\nabla_y G(x,y;\zeta^{(q,j)})}{d[y]^{\circ p-1}} \right).$$

*Under Theorems 3.2 to 3.4, we have*

$$\left\| \mathbb{E}_{\bar{\xi}}[H] - \left[ \frac{d\nabla_y g(x, y^*(x))}{d[y^*(x)]^{\circ p-1}} \right]^{-1} \right\| \leq \frac{1}{\mu} \left( 1 - \frac{\mu}{C} \right)^Q.$$

*Proof of Theorem B.5.* We follow a similar proof as (Ghadimi & Wang, 2018, Lemma 3.2). We have

$$\left\| \mathbb{E}_{\bar{\xi}}[H] - \left[ \frac{d\nabla_y g(x, y^*(x))}{d[y^*(x)]^{\circ p-1}} \right]^{-1} \right\| \leq \frac{1}{C} \left\| \sum_{q=Q}^{\infty} \left( I - \frac{1}{C} \frac{d\nabla_y G(x,y;\zeta^{(q,j)})}{d[y]^{\circ p-1}} \right)^q \right\|$$

$$\leq \frac{1}{C} \sum_{q=Q}^{\infty} \left\| \left( I - \frac{1}{C} \frac{d\nabla_y G(x,y;\zeta^{(q,j)})}{d[y]^{\circ p-1}} \right)^q \right\| \leq \frac{1}{\mu} \left( 1 - \frac{\mu}{C} \right)^Q,$$

where the second inequality uses triangle inequality, and the last inequality is due to Theorem B.2. □

**Remark**: Lemma B.4 provides the bias of the hypergradient due to the inaccurate estimate of the lower-level variable. This lemma is useful for the algorithm design and analysis in Section 5. Also, in Section 5, we analyze the bias and variance of the estimated hypergradient $\hat{\nabla} f(x, y, \bar{\xi})$ induced by Neumann series and Algorithm 1 and 2.

## B.7 SUFFICIENT AND NECESSARY CONDITION FOR THE DIFFERENTIABLITY ASSUMPTION

**Lemma B.6** (Sufficient And Necessary Condition For the Differentiablity Assumption). *Fix $p \geq 2$ and set $\alpha := \frac{1}{p-1} \in (0, 1)$. Define the sign–preserving, coordinatewise power map $S_\alpha : \mathbb{R}^d \to \mathbb{R}^d$ by $S_\alpha(z) = \text{sgn}(z) \odot |z|^\alpha$ so that $z_i = \text{sgn}(y_i) |y_i|^{p-1}$ where $y = S_\alpha(z)$. Let $h : \mathbb{R}^d \to \mathbb{R}$ be differentiable near 0 and define $r(z) := h(S_\alpha(z))$. Then $r(z)$ is differentiable at $z = 0$ with $\nabla r(0) = 0$ if and only if*

$$\lim_{y \to 0} \frac{\|\nabla h(y)\|}{\|y\|^{p-2}} = 0.$$

*Proof.* By definition, $r$ is differentiable at 0 with $\nabla r(0) = 0$ iff $\lim_{z \to 0} \frac{|r(z)-r(0)|}{\|z\|} = 0$.

Let $y = S_\alpha(z)$. Then

$$\|z\| = \left( \sum_{i=1}^{d} |y_i|^{2(p-1)} \right)^{1/2}.$$

**(Sufficiency).** Suppose $\lim_{y \to 0} \frac{\|\nabla h(y)\|}{\|y\|^{p-2}} = 0$. Since $h$ is differentiable, for each $y$ there exists $\xi$ on the line from 0 to $y$ such that $h(y) - h(0) = \nabla h(\xi)^\top y$. Hence

$$\frac{|r(z) - r(0)|}{\|z\|} = \frac{|h(y) - h(0)|}{\|z\|} \leq \|\nabla h(\xi)\| \frac{\|y\|}{\|z\|}.$$

Define $M := \max_i |y_i|$, we have $\|y\| \leq \sqrt{d} M$ and $\|z\| \geq M^{p-1}$, so

$$\frac{\|y\|}{\|z\|} \leq \sqrt{d} M^{-(p-2)} \leq (\sqrt{d})^{p-1} \|y\|^{-(p-2)}.$$

Therefore

$$\limsup_{z \to 0} \frac{|r(z) - r(0)|}{\|z\|} \leq (\sqrt{d})^{p-1} \limsup_{y \to 0} \frac{\|\nabla h(y)\|}{\|y\|^{p-2}} = 0.$$

Thus $r$ is differentiable at 0 with $\nabla r(0) = 0$.

**(Necessity).** Conversely, assume $\lim_{z \to 0} \frac{|r(z) - r(0)|}{\|z\|} = 0$. By a standard result in calculus, we have

$$\frac{|r(z) - r(0)|}{\|z\|} = \left| \int_0^1 \nabla h(ty)^\top \frac{y}{\|z\|} dt \right| \geq \frac{1}{\sqrt{d}} \left( \int_0^1 \|\nabla h(ty)\| dt \right) \|y\|^{-(p-2)},$$

where we used $\|z\| = (\sum |y_i|^{2(p-1)})^{1/2} \leq \sqrt{d} \|y\|^{p-1}$.

Since $y = S_\alpha(z)$ is continuous in $z$, $z \to 0$ iff $y \to 0$. Hence taking $\liminf_{z \to 0}$ is equivalent to taking $\liminf_{y \to 0}$.

Taking $\liminf_{y \to 0}$ yields

$$0 \geq \frac{1}{\sqrt{d}} \liminf_{y \to 0} \frac{\|\nabla h(y)\|}{\|y\|^{p-2}}.$$

Since the ratio is nonnegative, it follows that $\lim_{y \to 0} \frac{\|\nabla h(y)\|}{\|y\|^{p-2}} = 0$.

Finally, away from the origin, $S_\alpha$ is differentiable with Jacobian

$$DS_\alpha(z) = \mathrm{diag} \left( \alpha |z_i|^{\alpha-1} \right)_{i=1}^d,$$

so for $z \neq 0$, the chain rule gives $\nabla r(z) = DS_\alpha(z)^\top \nabla h(S_\alpha(z))$. $\qquad \square$

### B.8    OTHER USEFUL LEMMAS

**Lemma B.7** (Variance). *Under Theorems 3.2 to 3.4, we have*

$$\mathbb{E}_{\bar{\xi}} \|\hat{\nabla} f(x, y; \bar{\xi}) - \mathbb{E}_{\bar{\xi}}[\hat{\nabla} f(x, y; \bar{\xi})]\|^2 \leq \sigma_1^2, \qquad where \quad \sigma_1^2 = \sigma_f^2 + \frac{3}{\mu^2} \left[ (\sigma_f^2 + l_{f,0}^2)(\sigma_{g,2}^2 + 2l_{g,1}^2) + \sigma_f^2 l_{g,1}^2 \right].$$

*Proof of Theorem B.7.* Following the proof of (Hong et al., 2023, Lemma 1) gives the result. $\qquad \square$

## C    PROOFS OF SECTION 5.3

### C.1    CONVERGENCE GUARANTEE FOR MINIMIZING SINGLE-LEVEL UNIFORMLY CONVEX FUNCTIONS

In this section we consider the problem of minimizing single-level objective function $\psi : \mathbb{R}^d \to \mathbb{R}$:

$$\min_{w \in \mathbb{R}^d} \psi(w). \tag{46}$$

Denote $w^* = \arg\min_{w \in \mathbb{R}^d} \psi(w)$ as the minimizer of $\psi$. Assume that we access $\nabla\psi(w)$ through an unbiased stochastic oracle, i.e., $\mathbb{E}_\pi[\nabla\psi(w; \pi)] = \nabla\psi(w)$. We rely on the following assumption for analysis in this section.

**Assumption C.1.** Assume function $\psi$ is $(\mu, p)$-uniformly convex (see Theorem 3.2). In addition, the noise satisfies $\mathbb{E}_\pi[\exp(\|\nabla\psi(w; \pi) - \nabla\psi(w)\|^2 / \sigma^2)] \leq \exp(1)$.

**Lemma C.2.** *Under Theorem C.1, if there exists a constant $G$ such that $\|\nabla\psi(x)\| \leq G$, then we have*

$$\psi(x) - \psi(x^*) \leq G(pG/\mu)^{\frac{1}{p-1}}.$$

*Proof of Theorem C.2.* By convexity of $\psi$ and the Cauchy-Schwarz inequality, we have

$$\psi(x) - \psi(x^*) \leq \langle \nabla\psi(x), x - x^* \rangle \leq G\|x - x^*\|.$$

By $(\mu, p)$-uniform convexity of $\psi$,

$$\psi(x) - \psi(x^*) \geq \frac{\mu}{p} \|x - x^*\|^p.$$

Combing the above inequalities together gives $\|x - x^*\| \leq (pG/\mu)^{\frac{1}{p-1}}$. Therefore,

$$\psi(x) - \psi(x^*) \leq G\|x - x^*\| \leq G(pG/\mu)^{\frac{1}{p-1}}.$$

$\square$

**Lemma C.3.** *Under Theorem C.1, for any given $w^*$, let $D$ be an upper bound on $\|w_1 - w^*\|$ and assume there exists a constant $G$ such that $\|\nabla\psi(w)\| \leq G$. Apply the update*

$$w_{t+1} = w_t - \gamma\nabla\psi(w_t; \pi_t)$$

*for $T$ iterations. Then for any $\delta \in (0, 1)$, with probability at least $1 - \delta$ we have*

$$\frac{1}{T}\sum_{t=1}^{T}\psi(w_t) - \psi(w^*) \leq 2\gamma(G^2 + \sigma^2)\log(2/\delta) + \frac{\|w_1 - w^*\|^2}{2\gamma T} + \frac{8(G + \sigma)D\sqrt{3\log(2/\delta)}}{\sqrt{T}}.$$

*Proof of Theorem C.3.* Define the filtration as $\mathcal{H}_t := \sigma(\pi_1, \ldots, \pi_{t-1})$, where $\sigma(\cdot)$ denotes the $\sigma$-algebra. With a minor abuse of notation, we use $\mathbb{E}_t[\cdot] = \mathbb{E}[\cdot \mid \mathcal{H}_t]$. By Theorem C.1, we have

$$\mathbb{E}_t\left[\exp\left(\frac{\|\nabla\psi(w_t; \pi_t)\|^2}{4G^2 + 4\sigma^2}\right)\right] \leq \mathbb{E}_t\left[\exp\left(\frac{\|\nabla\psi(w_t)\|^2 + \|\nabla\psi(w_t; \pi_t) - \nabla\psi(w_t)\|^2}{2G^2 + 2\sigma^2}\right)\right]$$

$$\leq \exp\left(\frac{1}{2}\right)\sqrt{\mathbb{E}_t\left[\exp\left(\frac{\|\nabla\psi(w_t; \pi_t) - \nabla\psi(w_t)\|^2}{G^2 + \sigma^2}\right)\right]} \leq \exp(1),$$

(47)

where the first inequality uses Young's inequality, the second inequality is due to Jensen's inequality. Since $\mathbb{E}_t[\langle\nabla\psi(w_t; \pi_t), w_t - w^*\rangle] = \langle\nabla\psi(w_t), w_t - w^*\rangle$, then

$$X_t := \langle\nabla\psi(w_t), w_t - w^*\rangle - \langle\nabla\psi(w_t; \pi_t), w_t - w^*\rangle$$

is a martingale difference sequence. Note that $|X_t|$ can be bounded as

$$|X_t| \leq \|\nabla\psi(w_t)\|\|w_t - w^*\| + \|\nabla\psi(w_t; \pi_t)\|\|w_t - w^*\| \leq 2GD + 2D\|\nabla\psi(w_t; \pi_t)\|,$$

where the last inequality uses $\|w_t - w^*\| \leq \|w_t - w_1\| + \|w_1 - w^*\| \leq 2D$ since $x_t, x^* \in \mathcal{B}(w_1, D)$. This implies that

$$\mathbb{E}_t\left[\exp\left(\frac{X_t^2}{64(G^2 + \sigma^2)D^2}\right)\right] \leq \mathbb{E}_t\left[\exp\left(\frac{4D^2(2G^2 + 2\|\nabla\psi(w_t; \pi_t)\|^2)}{64(G^2 + \sigma^2)D^2}\right)\right]$$

$$\leq \exp\left(\frac{1}{8}\right)\sqrt{\mathbb{E}_t\left[\exp\left(\frac{\|\nabla\psi(w_t; \pi_t)\|^2}{4G^2 + 4\sigma^2}\right)\right]} \leq \exp(1),$$

where the first inequality uses Young's inequality, the second inequality is due to Jensen's inequality, and the last inequality uses Eq. (47). By Theorem C.7, with probability at least $1 - \delta/2$, we have $\sum_{t=1}^{T}X_t \leq 8(G + \sigma)D\sqrt{3T\log(2/\delta)}$, which implies

$$\frac{1}{T}\sum_{t=1}^{T}\langle\nabla\psi(w_t), w_t - w^*\rangle - \langle\nabla\psi(w_t; \pi_t), w_t - w^*\rangle \leq \frac{8(G + \sigma)D\sqrt{3\log(2/\delta)}}{\sqrt{T}}. \quad (48)$$

Next,

$$\mathbb{E}\left[\exp\left(\frac{\sum_{t=1}^{T}\|\nabla\psi(w_t; \pi_t)\|^2}{4G^2 + 4\sigma^2}\right)\right] = \mathbb{E}\left[\mathbb{E}_T\left[\exp\left(\frac{\sum_{t=1}^{T}\|\nabla\psi(w_t; \pi_t)\|^2}{4G^2 + 4\sigma^2}\right)\right]\right]$$

$$= \mathbb{E}\left[\exp\left(\frac{\sum_{t=1}^{T-1}\|\nabla\psi(w_t; \pi_t)\|^2}{4G^2 + 4\sigma^2}\right)\mathbb{E}_T\left[\exp\left(\frac{\|\nabla\psi(w_T; \pi_T)\|^2}{4G^2 + 4\sigma^2}\right)\right]\right]$$

$$= \mathbb{E}\left[\exp\left(\frac{\sum_{t=1}^{T-1}\|\nabla\psi(w_t; \pi_t)\|^2}{4G^2 + 4\sigma^2}\right)\cdot\exp(1)\right],$$

where the last inequality uses Eq. (47). Apply the above procedure inductively, we obtain

$$\mathbb{E}\left[\exp\left(\frac{\sum_{t=1}^{T}\|\nabla\psi(w_t;\pi_t)\|^2}{4G^2+4\sigma^2}\right)\right] \le \exp(T).$$

By Markov's inequality, with probability at least $1 - \delta/2$, we have

$$\sum_{t=1}^{T}\|\nabla\psi(w_t;\pi_t)\|^2 \le 4(G^2+\sigma^2)T\log(2/\delta).$$

By Theorem C.6 and Eq. (48), we conclude that

$$\frac{1}{T}\sum_{t=1}^{T}\psi(w_t) - \psi(w^*) \le 2\gamma(G^2+\sigma^2)\log(2/\delta) + \frac{\|w_1-w^*\|^2}{2\gamma T} + \frac{8(G+\sigma)D\sqrt{3\log(2/\delta)}}{\sqrt{T}}.$$

$\square$

**Lemma C.4.** *Define $\Delta_k$ and $V_k$, choose $\gamma_1$ and $T_1$ as*

$$\Delta_k = \psi(w_k) - \psi(w^*), \quad V_k = \frac{G(pG/\mu)^{\frac{1}{p-1}}}{2^{k-1}} \quad and \quad \gamma_1 = \frac{G(pG/\mu)^{\frac{1}{p-1}}}{24(G^2+\sigma^2)}, \quad T_1 = \frac{60^2(G^2+\sigma^2)}{G^2}.$$

(49)

*For any $k$, with probability at least $(1-\tilde{\delta})^{k-1}$ we have $\Delta_k \le V_k\log(2/\tilde{\delta})$.*

*Proof of Theorem C.4.* Denote $\iota := \log(2/\tilde{\delta})$. We will prove the lemma by induction on $k$, i.e., $\Delta_k \le V_k\iota$.

**Base Case.** The claim is true for $k = 1$ since $\Delta_1 \le V_1\iota$ by Theorem C.2.

**Induction.** Assume that $\Delta_k \le V_k\iota$ for some $k \ge 1$ with probability at least $(1-\tilde{\delta})^{k-1}$ and now we prove the claim for $k+1$. Since $\Delta_k \ge \frac{\mu}{p}\|w_1^k - w^*\|^p$ by $(\mu,p)$-uniform convexity, which, combined with the induction hypothesis $\Delta_k \le V_k\iota$ implies that

$$\|w_1^k - w^*\| \le (p\Delta_k/\mu)^{\frac{1}{p}} = D_k.$$

(50)

Apply Theorem C.3 with $D = D_k$ and hence with probability at least $1 - \tilde{\delta}$,

$$\begin{aligned}
\Delta_{k+1} &= \psi(w_1^{k+1}) - \psi(w^*) \\
&\le 2\gamma_k(G^2+\sigma^2)\iota + \frac{\|w_1^k-w^*\|^2}{2\gamma_k T_k} + \frac{8(G+\sigma)D_k\sqrt{3\iota}}{\sqrt{T_k}} \\
&\le 2\gamma_k(G^2+\sigma^2)\iota + \frac{(p\Delta_k/\mu)^{\frac{2}{p}}}{2\gamma_k T_k} + \frac{20(G+\sigma)(p\Delta_k/\mu)^{\frac{1}{p}}\sqrt{\iota}}{\sqrt{T_k}} \\
&\le \frac{\gamma_1(G^2+\sigma^2)\iota}{2^{k-2}} + \frac{(pV_k\iota/\mu)^{\frac{2}{p}}}{2\gamma_1 T_1 \cdot 2^{\frac{p-2}{p}(k-1)}} + \frac{20(G+\sigma)(pV_k\iota/\mu)^{\frac{1}{p}}\sqrt{\iota}}{\sqrt{T_1 2^{\tau(k-1)}}} \\
&\le \frac{V_k\iota}{12} + \frac{V_k\iota}{300} + \frac{V_k\iota}{3} \\
&\le \frac{V_k\iota}{2} = V_{k+1}\iota,
\end{aligned}$$

where the first inequality uses Theorem C.3, the second inequality is due to Eq. (50), the third inequality uses the induction hypothesis and the definition of $\gamma_k$ and $T_k$, and the fourth inequality is due to the choice of $\gamma_1$ and $T_1$ as in Eq. (50).

Factoring in the conditioned event $\Delta_k \le V_k\iota$, which happens with probability at least $(1-\tilde{\delta})^{k-1}$, thus we obtain that $\Delta_{k+1} \le V_{k+1}\iota$ with probability at least $(1-\tilde{\delta})^k$. $\square$

**Theorem C.5.** *Under Theorem C.1, given any* $\delta \in (0,1)$, *set* $\tilde{\delta} = \delta/k^\dagger$ *for* $k^\dagger = \lfloor \frac{1}{\tau} \log_2((\frac{T}{T_1})(2^\tau - 1) + 1) \rfloor$. *Set the parameters* $\gamma_1$, $T_1$ *and* $D_1$ *as*

$$\gamma_1 = \frac{G(pG/\mu)^{\frac{1}{p-1}}}{24(G^2 + \sigma^2)}, \quad T_1 = \frac{60^2(G^2 + \sigma^2)}{G^2}, \quad D_1 = \min\left\{ \left(\frac{pG}{\mu}\right)^{\frac{1}{p-1}} \log(2/\tilde{\delta}), \|w_1^1 - w^*\| \right\}$$

(51)

*in Algorithm 1. Then with probability at least* $1 - \delta$, *we have*

$$\psi(w_1^k) - \psi(w^*) \leq \frac{(60^2(G^2 + \sigma^2))^{\frac{p}{2(p-1)}}(p/\mu)^{\frac{1}{p-1}} \log(2/\tilde{\delta})}{T^{\frac{p}{2(p-1)}}} = O\left(T^{-\frac{p}{2(p-1)}}\right),$$

$$\|w_1^k - w^*\| \leq \frac{(60^2(G^2 + \sigma^2))^{\frac{1}{2(p-1)}}(p/\mu)^{\frac{1}{p-1}} \log(2/\tilde{\delta})}{T^{\frac{1}{2(p-1)}}} = O\left(T^{-\frac{1}{2(p-1)}}\right).$$

*Proof of Theorem C.5.* Recall $\tau = 2(p-1)/p$ as defined in Algorithm 1. By Theorem C.4, with probability at least $1 - \tilde{\delta}$,

$$\psi(w_1^{k^\dagger+1}) - \psi(w^*) = \Delta_{k^\dagger+1} \leq V_{k^\dagger+1} \log(2/\tilde{\delta})$$

$$= \frac{G(pG/\mu)^{\frac{1}{p-1}} \log(2/\tilde{\delta})}{2^{k^\dagger}} \leq G(pG/\mu)^{\frac{1}{p-1}} \left(\left(\frac{T}{T_1}\right)(2^\tau - 1) + 1\right)^{-\frac{1}{\tau}} \log(2/\tilde{\delta})$$

$$\leq \frac{T_1^{\frac{1}{\tau}} G(pG/\mu)^{\frac{1}{p-1}} \log(2/\tilde{\delta})}{T^{\frac{1}{\tau}}} = \frac{(60^2(G^2 + \sigma^2))^{\frac{p}{2(p-1)}}(p/\mu)^{\frac{1}{p-1}} \log(2/\tilde{\delta})}{T^{\frac{p}{2(p-1)}}},$$

where the second inequality uses the definition of $k^\dagger$, the third inequality is due to $\tau \geq 1$, and the last equality uses the definition of $\tau$ and the choice of $T_1$ as in Eq. (51). Also, by $(\mu, p)$-uniform convexity of $\psi$ we have

$$\psi(w_1^{k^\dagger+1}) - \psi(w^*) \geq \frac{\mu}{p}\|w_1^{k^\dagger+1} - w^*\|^p.$$

Combing the above inequalities yields the results. $\qquad\square$

**Lemma C.6** ((Hazan & Kale, 2014, Lemma 6)). *Starting from an arbitrary point* $w_1 \in \mathbb{R}^d$, *apply* $T$ *iterations of the update*

$$w_{t+1} = w_t - \gamma \nabla \psi(w_t; \pi_t).$$

*Then for any point* $w^* \in \mathbb{R}^d$, *we have*

$$\sum_{t=1}^{T} \langle \nabla \psi(w_t; \pi_t), w_t - w^* \rangle \leq \frac{\gamma}{2} \sum_{t=1}^{T} \|\nabla \psi(w_t; \pi_t)\|^2 + \frac{\|w_1 - w^*\|^2}{2\gamma}.$$

**Lemma C.7** ((Hazan & Kale, 2014, Lemma 14)). *Let* $X_1, \ldots, X_T$ *be a martingale difference sequence, i.e.,* $\mathbb{E}_t[X_t] = 0$ *for all* $t$. *Suppose that there exists* $\sigma_1, \ldots, \sigma_T$ *such that* $\mathbb{E}_t[\exp(X_t^2/\sigma_t^2)] \leq \exp(1)$. *Then with probability at least* $1 - \delta$, *we have*

$$\sum_{t=1}^{T} X_t \leq \sqrt{3 \log(1/\delta) \sum_{t=1}^{T} \sigma_t^2}.$$

## C.2 PROOF OF LEMMA 5.2

We will use a short hand $y^* = y^*(x)$.

**Lemma C.8.** *Under Theorem 3.2, if* $y^* \in \mathcal{B}(y; R)$ *for some* $R > 0$, *then for all* $\bar{y} \in \mathcal{B}(y; R)$,

$$\|\nabla_y g(x, \bar{y})\| \leq \frac{(2^{(2L_1 R+1)} - 1)L_1}{L_0}.$$

*Proof of Theorem C.8.* For any $\bar{y} \in \mathcal{B}(y; R)$, let $y_0' = y^*$ and $y_j' = \bar{y}$, then there exists $y_0', y_1', \ldots, y_j'$ with $j = \lceil L_1 \|\bar{y} - y^*\| \rceil$ such that $\|y_i' - y_{i-1}'\| \leq 1/L_1$ for $i = 1, \ldots, j$. We will prove $\|\nabla_y g(x, y_i')\| \leq (2^i - 1)L_1/L_0$ for all $i \leq j$ by induction.

**Base Case.** For $y_1'$, by Theorem 3.2 we have

$$\|\nabla_y g(x, y_1') - \nabla_y g(x, y_0')\| \leq (L_0 + L_1 \|\nabla_y g(x, y_0')\|)\|y_1' - y_0'\| \leq \frac{L_0}{L_1},$$

where the last inequality uses $y_0' = y^*$. This implies that $\|\nabla_y g(x, y_1')\| \leq L_0/L_1$.

**Induction.** Assume that $\|\nabla_y g(x, y_i')\| \leq (2^i - 1)L_0/L_1$ holds for some $i \leq j - 1$. Then for $y_{i+1}'$ we have

$$\|\nabla_y g(x, y_{i+1}') - \nabla_y g(x, y_i')\| \leq (L_0 + L_1 \|\nabla_y g(x, y_i')\|)\|y_{i+1}' - y_i'\| \leq \frac{2^i L_0}{L_1},$$

where the last inequality uses the induction hypothesis. By triangle inequality and the induction hypothesis we obtain $\|\nabla_y g(x, y_{i+1}')\| \leq (2^{i+1} - 1)L_1/L_0$. Therefore, we conclude that for any $\bar{y} \in \mathcal{B}(y; R)$,

$$\|\nabla_y g(x, \bar{y})\| \leq \frac{(2^j - 1)L_1}{L_0} = \frac{(2^{\lceil L_1 \|\bar{y} - y^*\| \rceil} - 1)L_1}{L_0} \leq \frac{(2^{(2L_1 R + 1)} - 1)L_1}{L_0},$$

where the last inequality uses $\|\bar{y} - y^*\| \leq 2R$ since $\bar{y}, y^* \in \mathcal{B}(y; R)$. $\qquad\square$

**Lemma C.9** (Restatement of Theorem 5.2). *For any given $\delta \in (0, 1)$ and $\epsilon > 0$, set $\tilde{\delta} = \delta/(Tk^\dagger)$ for $k^\dagger = \lfloor \frac{1}{\tau} \log_2((\frac{K_t}{K_{t,1}})(2^\tau - 1) + 1) \rfloor$, where $\tau = 2(p-1)/p$ is defined in Algorithm 1. Choose $\{\alpha_{t,1}\}, \{K_{t,1}\}, \{R_{t,1}\}, \{K_t\}$ as*

$$G_t = \begin{cases} (2^{(2L_1 \|y_0 - y_0^*\| + 1)} - 1)\frac{L_1}{L_0} & t = 0 \\ \frac{L_1}{L_0} & t \geq 1 \end{cases}, \quad R_{t,1} = \begin{cases} \min\left\{ (pG_t/\mu)^{\frac{1}{p-1}} \log(2/\tilde{\delta}), \|y_0 - y_0^*\| \right\} & t = 0 \\ \min\left\{ \frac{\epsilon}{4L_{\phi_2}}, \frac{1}{L_1} \right\} & t \geq 1 \end{cases},$$

$$\tag{52}$$

$$\alpha_{t,1} = \frac{G_t (pG_t/\mu)^{\frac{1}{p-1}}}{24(G_t^2 + \sigma_{g,1}^2)}, \quad K_{t,1} = \frac{60^2 (G_t^2 + \sigma_{g,1}^2)}{G_t^2}, \quad K_t = \frac{60^2 (G_t^2 + \sigma_{g,1}^2)(p/\mu)^2 (\log(2/\tilde{\delta}))^{2(p-1)}}{(\min\{\epsilon/8L_{\phi_2}, 1/2L_1\})^{2(p-1)}}.$$

$$\tag{53}$$

*For any sequence $\{\tilde{x}_t\}$ such that $\tilde{x}_0 = x_0$ and $\|\tilde{x}_{t+1} - \tilde{x}_t\| = \eta$ for $\eta$ satisfying*

$$\eta \leq \left( \frac{1}{Il_p} \min\left\{ \frac{\epsilon}{8L_{\phi_2}}, \frac{1}{2L_1} \right\} \right)^{p-1}, \tag{54}$$

*let $\{\tilde{y}_t\}$ be the output produced by Algorithm 2. Then with probability at least $1 - \delta$, for all $t \in [T]$ we have $\|\tilde{y}_t - \tilde{y}_t^*\| \leq \min\{\epsilon/4L_{\phi_2}, 1/L_1\}$.*

*Proof of Theorem C.9.* For $t = 0$, by Theorems C.5 and C.8 and the choices of $\alpha_{0,1}, K_{0,1}, R_{0,1}$ as in Eq. (52) and Eq. (53), with probability at least $1 - \delta/T$ we have $\|\tilde{y}_1 - \tilde{y}_0^*\| \leq \min\{\epsilon/8L_{\phi_2}, 1/2L_1\}$. For $1 \leq t \leq I$, we have

$$\|\tilde{y}_t - \tilde{y}_t^*\| = \|\tilde{y}_1 - \tilde{y}_t^*\| \leq \|\tilde{y}_1 - \tilde{y}_0^*\| + \sum_{i=1}^t \|\tilde{y}_{i-1}^* - \tilde{y}_i^*\| \leq \min\{\epsilon/8L_{\phi_2}, 1/2L_1\} + Il_p \|\tilde{x}_{i-1} - \tilde{x}_i\|^{\frac{1}{p-1}}$$

$$= \min\{\epsilon/8L_{\phi_2}, 1/2L_1\} + Il_p \eta^{\frac{1}{p-1}} \leq \min\{\epsilon/4L_{\phi_2}, 1/L_1\},$$

where the first inequality uses triangle inequality, the second inequality is due to $t \leq I$ and Theorem 4.2, the last inequality uses the choice of $\eta$ as in Eq. (54). For $t \geq I$, apply Theorems C.5 and C.8 with the choices of $\alpha_{t,1}, K_{t,1}, R_{t,1}$ as in Eq. (52) and Eq. (53), then follow the above procedure inductively, we obtain with probability at least $1 - \delta$ that for all $t$, $\|\tilde{y}_t - \tilde{y}_t^*\| \leq \min\{\epsilon/4L_{\phi_2}, 1/L_1\}$. $\qquad\square$

## C.3 PROOF OF LEMMA 5.3

**Corollary C.10** (Restatement of Theorem 5.3). *Let $\{x_t\}$ and $\{y_t\}$ be the iterates generated by Algorithm 2. For any given $\delta \in (0,1)$ and $\epsilon > 0$, under the same parameter setting in Theorem C.9, with probability at least $1 - \delta$ (denote this event as $\mathcal{E}$) we have $\|y_t - y_t^*\| \leq \min\{\epsilon/4L_{\phi_2}, 1/L_1\}$ for all $t \geq 1$.*

*Proof of Theorem C.10.* By line 8 of Algorithm 2, we have $\|x_{t+1} - x_t\| = \eta$. Setting $\{\tilde{x}_t\} = \{x_t\}$ yields the result. $\square$

## C.4 PROOF OF LEMMA 5.4

**Lemma C.11.** *Under Theorems 3.2 and 3.3, define $\epsilon_t := m_t - \nabla\Phi(x_t)$, then we have*

$$\Phi(x_{t+1}) \leq \Phi(x_t) - \eta\|\nabla\Phi(x_t)\| + 2\eta\|\epsilon_t\| + \frac{(p-1)L_{\phi_1}}{p}\eta^{\frac{p}{p-1}} + \frac{L_{\phi_2}}{2}\eta^2.$$

*Furthermore,*

$$\sum_{t=1}^{T}\|\nabla\Phi(x_t)\| \leq \frac{\Delta_\phi}{\eta} + T\left(\frac{(p-1)L_{\phi_1}}{p}\eta^{\frac{1}{p-1}} + \frac{L_{\phi_2}}{2}\eta\right) + 2\sum_{t=1}^{T}\|\epsilon_t\|.$$

*Proof of Theorem C.11.* By Theorem 4.1, we have

$$
\begin{aligned}
\Phi(x_{t+1}) &\leq \Phi(x_t) + \langle\nabla\Phi(x_t), x_{t+1} - x_t\rangle + \frac{(p-1)L_{\phi_1}}{p}\|x_{t+1} - x_t\|^{\frac{p}{p-1}} + \frac{L_{\phi_2}}{2}\|x_{t+1} - x_t\|^2 \\
&= \Phi(x_t) - \eta\left\langle m_t - \epsilon_t, \frac{m_t}{\|m_t\|}\right\rangle + \frac{(p-1)L_{\phi_1}}{p}\eta^{\frac{p}{p-1}} + \frac{L_{\phi_2}}{2}\eta^2 \\
&= \Phi(x_t) - \eta\|m_t\| + \eta\left\langle\epsilon_t, \frac{m_t}{\|m_t\|}\right\rangle + \frac{(p-1)L_{\phi_1}}{p}\eta^{\frac{p}{p-1}} + \frac{L_{\phi_2}}{2}\eta^2 \\
&\leq \Phi(x_t) - \eta\|\nabla\Phi(x_t) + \epsilon_t\| + \eta\|\epsilon_t\| + \frac{(p-1)L_{\phi_1}}{p}\eta^{\frac{p}{p-1}} + \frac{L_{\phi_2}}{2}\eta^2 \\
&\leq \Phi(x_t) - \eta\|\nabla\Phi(x_t)\| + 2\eta\|\epsilon_t\| + \frac{(p-1)L_{\phi_1}}{p}\eta^{\frac{p}{p-1}} + \frac{L_{\phi_2}}{2}\eta^2,
\end{aligned}
\tag{55}
$$

where the first equality uses the update rule (line 8) of Algorithm 2, the second inequality is due to Cauchy–Schwarz inequality, and the last inequality uses triangle inequality. Rearranging Eq. (55) and taking summation yields the result. $\square$

**Lemma C.12** (Restatement of Theorem 5.4). *Under Theorems 3.2 to 3.4 and event $\mathcal{E}$, we have*

$$\sum_{t=1}^{T}\mathbb{E}\|\epsilon_t\| \leq \frac{\sigma_1}{1-\beta} + T\sqrt{1-\beta}\sigma_1 + TL_{\phi_2}\min\left\{\frac{\epsilon}{4L_{\phi_2}}, \frac{1}{L_1}\right\} + \frac{Tl_{g,1}l_{f,0}}{\mu}\left(1 - \frac{\mu}{C}\right)^Q + \frac{T}{1-\beta}\left(L_{\phi_1}\eta^{\frac{1}{p-1}} + L_{\phi_2}\eta\right).$$

*Proof of Theorem C.12.* Define $\hat{\epsilon}_t = \hat{\nabla}f(x_t, y_t; \bar{\xi}_t) - \nabla\Phi(x_t)$ and $S(a,b) = \nabla\Phi(a) - \nabla\Phi(b)$. By Theorem 4.1, we have

$$\|S(x_t, x_{t+1})\| = \|\Phi(x_t) - \Phi(x_{t+1})\| \leq L_{\phi_1}\|x_t - x_{t+1}\|^{\frac{1}{p-1}} + L_{\phi_2}\|x_t - x_{t+1}\| \leq L_{\phi_1}\eta^{\frac{1}{p-1}} + L_{\phi_2}\eta.
\tag{56}$$

For all $t \geq 1$, we have the following recursion:

$$\epsilon_{t+1} = \beta\epsilon_t + (1-\beta)\hat{\epsilon}_{t+1} + \beta S(x_t, x_{t+1}).
\tag{57}$$

Unrolling the recursion gives

$$\epsilon_{t+1} = \beta^t\epsilon_1 + (1-\beta)\sum_{i=0}^{t-1}\beta^i\hat{\epsilon}_{t+1-i} + \beta\sum_{i=0}^{t-1}\beta^i S(x_{t-i}, x_{t+1-i}).$$

By triangle inequality and Eq. (56), we have

$$
\begin{aligned}
\|\epsilon_{t+1}\| &\leq \beta^t \|\epsilon_1\| + (1-\beta) \left\| \sum_{i=0}^{t-1} \beta^i \hat{\epsilon}_{t+1-i} \right\| + \beta \left( L_{\phi_1} \eta^{\frac{1}{p-1}} + L_{\phi_2} \eta \right) \sum_{i=0}^{t-1} \beta^i \\
&\leq \beta^t \underbrace{\|\epsilon_1\|}_{(A)} + (1-\beta) \underbrace{\left\| \sum_{i=0}^{t-1} \beta^i \hat{\epsilon}_{t+1-i} \right\|}_{(B)} + \frac{\beta}{1-\beta} \left( L_{\phi_1} \eta^{\frac{1}{p-1}} + L_{\phi_2} \eta \right).
\end{aligned}
\tag{58}
$$

**Bounding** $(A)$**.** Observe that $\epsilon_1 = \hat{\epsilon}_1$. Taking expectation and using Jensen's inequality, we have

$$
\mathbb{E}\|\epsilon_1\| = \mathbb{E}\|\hat{\epsilon}_1\| \leq \sqrt{\mathbb{E}\|\hat{\epsilon}_1\|^2} \leq \sigma_1.
$$

**Bounding** $(B)$**.** By triangle inequality, we have

$$
\begin{aligned}
\mathbb{E} \left\| \sum_{i=0}^{t-1} \beta^i \hat{\epsilon}_{t+1-i} \right\| &\leq \mathbb{E} \left\| \sum_{i=0}^{t-1} \beta^i (\hat{\nabla} f(x_i, y_i; \bar{\xi}_i) - \mathbb{E}_t[\hat{\nabla} f(x_i, y_i; \bar{\xi}_i)]) \right\| + \mathbb{E} \left\| \sum_{i=0}^{t-1} \beta^i (\mathbb{E}_t[\hat{\nabla} f(x_i, y_i; \bar{\xi}_i)] - \nabla \Phi(x_i)) \right\| \\
&\leq \sqrt{\sum_{i=0}^{t-1} \beta^{2i} \mathbb{E}\|\hat{\nabla} f(x_i, y_i; \bar{\xi}_i) - \mathbb{E}_t[\hat{\nabla} f(x_i, y_i; \bar{\xi}_i)]\|^2} + \sum_{i=0}^{t-1} \beta^i \left( L_{\phi_2} \|y_i - y_i^*\| + \frac{l_{g,1} l_{f,0}}{\mu} \left(1 - \frac{\mu}{C}\right)^Q \right) \\
&\leq \frac{\sigma_1}{\sqrt{1-\beta}} + \frac{L_{\phi_2}}{1-\beta} \min\left\{ \frac{\epsilon}{4L_{\phi_2}}, \frac{1}{L_1} \right\} + \frac{l_{g,1} l_{f,0}}{\mu(1-\beta)} \left(1 - \frac{\mu}{C}\right)^Q,
\end{aligned}
$$

where the second inequality uses Jensen's inequality and the fact that for $i \neq j$, $\bar{\xi}_i$ and $\bar{\xi}_j$ are uncorrelated, and the last inequality is due to Theorem B.7 and Theorem C.10.

Returning to Eq. (58), we obtain

$$
\mathbb{E}\|\epsilon_{t+1}\| \leq \beta^t \sigma_1 + \sqrt{1-\beta}\, \sigma_1 + L_{\phi_2} \min\left\{ \frac{\epsilon}{4L_{\phi_2}}, \frac{1}{L_1} \right\} + \frac{l_{g,1} l_{f,0}}{\mu} \left(1 - \frac{\mu}{C}\right)^Q + \frac{\beta}{1-\beta} \left( L_{\phi_1} \eta^{\frac{1}{p-1}} + L_{\phi_2} \eta \right).
$$

Summing from $t = 1$ to $T$ yields

$$
\sum_{t=1}^{T} \mathbb{E}\|\epsilon_t\| \leq \frac{\sigma_1}{1-\beta} + T\sqrt{1-\beta}\, \sigma_1 + T L_{\phi_2} \min\left\{ \frac{\epsilon}{4L_{\phi_2}}, \frac{1}{L_1} \right\} + \frac{T l_{g,1} l_{f,0}}{\mu} \left(1 - \frac{\mu}{C}\right)^Q + \frac{T}{1-\beta} \left( L_{\phi_1} \eta^{\frac{1}{p-1}} + L_{\phi_2} \eta \right).
$$

$\square$

## D  PROOF OF MAIN THEOREM 5.1

**Theorem D.1** (Restatement of Theorem 5.1)**.** *Under Theorems 3.2 to 3.4, for any given $\delta \in (0,1)$ and $\epsilon > 0$, set $\tilde{\delta} = \delta/(Tk^\dagger)$ for $k^\dagger = \lfloor \frac{1}{\tau} \log_2((\frac{K_t}{K_{t,1}})(2^\tau - 1) + 1) \rfloor$, where $\tau = 2(p-1)/p$ is defined in Algorithm 1. Choose $\{\alpha_{t,1}\}, \{K_{t,1}\}, \{R_{t,1}\}, \{K_t\}$ as*

$$
G_t = \begin{cases} (2^{(2L_1\|y_0 - y_0^*\| + 1)} - 1)\dfrac{L_1}{L_0} & t = 0 \\[2mm] \dfrac{L_1}{L_0} & t \geq 1 \end{cases}, \quad R_{t,1} = \begin{cases} \min\left\{ (pG_t/\mu)^{\frac{1}{p-1}} \log(2/\tilde{\delta}), \|y_0 - y_0^*\| \right\} & t = 0 \\[2mm] \min\left\{ \dfrac{\epsilon}{L_{\phi_2}}, \dfrac{1}{L_1} \right\} & t \geq 1 \end{cases},
\tag{59}
$$

$$
\alpha_{t,1} = \frac{G_t (pG_t/\mu)^{\frac{1}{p-1}}}{24(G_t^2 + \sigma_{g,1}^2)}, \quad K_{t,1} = \frac{60^2(G_t^2 + \sigma_{g,1}^2)}{G_t^2}, \quad K_t = \frac{60^2(G_t^2 + \sigma_{g,1}^2)(p/\mu)^2 (\log(2/\tilde{\delta}))^{2(p-1)}}{(\min\{\epsilon/2L_{\phi_2}, 1/2L_1\})^{2(p-1)}}.
\tag{60}
$$

*In addition, choose $\beta, \eta, I$ and $Q$ as*

$$
1 - \beta = \min\left\{ 1, \frac{c_1 \epsilon^2}{\sigma_1^2} \right\}, \quad \eta = c_2 \min\left\{ \left( \epsilon \cdot \min\left\{ \frac{1-\beta}{L_{\phi_1}}, \frac{p}{(p-1)L_{\phi_1}}, \frac{1-\beta}{l_p L_{\phi_2}} \right\} \right)^{p-1}, \frac{(1-\beta)\epsilon}{L_{\phi_2}}, \frac{\epsilon}{L_{\phi_2}} \right\},
\tag{61}
$$

$$I = \frac{1}{1-\beta}, \quad Q = \ln\left(\frac{\mu\epsilon}{4l_{g,1}l_{f,0}}\right) \Big/ \ln\left(1 - \frac{\mu}{C}\right). \tag{62}$$

*Let* $T = \frac{C_1 \Delta_\phi}{\eta\epsilon}$. *Then with probability at least* $1 - \delta$ *over the randomness in* $\mathcal{F}_y$, *we have* $\frac{1}{T}\sum_{t=1}^{T}\mathbb{E}\|\nabla\Phi(x_t)\| \leq \epsilon$, *where the expectation is taken over the randomness in* $\mathcal{F}_{T+1}$. *The total oracle complexity is* $\widetilde{O}(\epsilon^{-5p+6})$.

*Proof of Theorem D.1.* We apply Theorems C.11 and C.12 to obtain that, under event $\mathcal{E}$,

$$\frac{1}{T}\sum_{t=1}^{T}\mathbb{E}\|\nabla\Phi(x_t)\| \leq \frac{\Delta_\phi}{T\eta} + \left(\frac{(p-1)L_{\phi_1}}{p}\eta^{\frac{1}{p-1}} + \frac{L_{\phi_2}}{2}\eta\right) + \frac{2}{T}\sum_{t=1}^{T}\mathbb{E}\|\epsilon_t\|$$

$$\leq \frac{\Delta_\phi}{T\eta} + \left(\frac{(p-1)L_{\phi_1}}{p}\eta^{\frac{1}{p-1}} + \frac{L_{\phi_2}}{2}\eta\right) + \frac{2\sigma_1}{T(1-\beta)} + 2\sqrt{1-\beta}\sigma_1 + 2L_{\phi_2}\min\left\{\frac{\epsilon}{4L_{\phi_2}}, \frac{1}{L_1}\right\}$$

$$+ \frac{2}{1-\beta}\left(L_{\phi_1}\eta^{\frac{1}{p-1}} + L_{\phi_2}\eta\right) + \frac{l_{g,1}l_{f,0}}{\mu}\left(1 - \frac{\mu}{C}\right)^Q$$

$$\leq \left(\frac{1}{C_1} + c_2^{\frac{1}{p-1}} + \frac{c_2}{2} + \frac{2c_2\sigma_1\epsilon}{C_1\Delta_\phi L_{\phi_2}} + 2\sqrt{c_1} + \frac{1}{2} + 2c_2^{\frac{1}{p-1}} + 2c_2 + \frac{1}{4}\right)\epsilon$$

$$\leq \epsilon,$$

where the third inequality uses the choice of $\eta$, $\beta$ and $Q$ as in Eq. (61) and Eq. (62), the last inequality is due to the choice of small enough constants $c_1, c_2$ and large enough constant $C_1$.

Moreover, the total oracle complexity is (assume target accuracy $\epsilon$ is small enough):

$$O\left(T + \sum_{j=0}^{\lceil T/I \rceil} K_j I Q\right) = \widetilde{O}\left(\epsilon^{-3p+2} + \epsilon^{-5p+6}\right) = \widetilde{O}\left(\epsilon^{-5p+6}\right). \tag{63}$$

$\square$

# E ADDITIONAL EXPERIMENTS

## E.1 MORE EXPERIMENTS FOR SYNTHETIC DATA

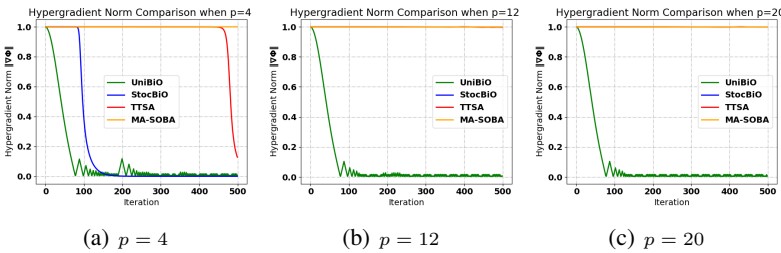

(a) $p = 4$        (b) $p = 12$        (c) $p = 20$

Figure 3: Results of bilevel optimization on the synthetic example 2 when $p = \{4, 12, 20\}$. All algorithms are initialized at $(x_0, y_0) = (0.001, 0.001)$, and the upper-level variable is updated for $T = 500$ iterations. The performance of the algorithms was evaluated through the ground-truth hypergradient given by $\nabla\Phi(x) = \sin(x)\cos(\sin(x))$. For all algorithms, learning rates are optimally tuned with a grid search over the range $[0.01, 1]$.

In this section, we conducted extensive synthetic experiments to rigorously compare UniBiO against prominent LLSC-based algorithms, including StocBiO (Ji et al., 2021), TTSA (Hong et al., 2023), and MA-SOBA (Chen et al., 2023), under a deterministic setting. All experiments were initialized at $(x_0, y_0) = (0.001, 0.001)$, with the upper-level iteration number fixed at $T = 500$. Algorithm performance was evaluated through the ground-truth hypergradient given by $\nabla\Phi(x) = \sin(x)\cos(\sin(x))$ across varying $p \in \{4, 12, 20\}$.

**Parameter Settings:** For UniBiO and StocBiO, we set Neumann series iterations as $Q = 10$. Momentum for UniBio and MA-SOBA was fixed at 0.9. The optimal upper- ($\eta_U L$) and lower-level learning rates ($\eta_{LL}$) for each algorithm were determined through a grid search over the range $[0.01, 1]$. Specifically the learning rates are: UniBiO ($\eta_{UL} = 0.02$, $\eta_{LL} = 1.0$); StocBiO ($\eta_{UL} = 0.5$, $\eta_{LL} = 0.1$); TTSA ($\eta_{UL} = 0.1$, $\eta_{LL} = 0.1$); MA-SOBA ($\eta_{UL} = 1.0$, $\eta_{LL} = 0.01$, $\eta_z = 0.01$). Other fixed parameters included: UniBio ($I = 10$, $N_1 = 5$, $D_1 = 1$, $T_y = 100$), StocBiO (the number of inner iterations $T_y = 5$), and MA-SOBA (the auxiliary variable $z$ is initialized at $z_0 = 0$).

### E.2 MORE EXPERIMENTS FOR DATA HYPER-CLEANING

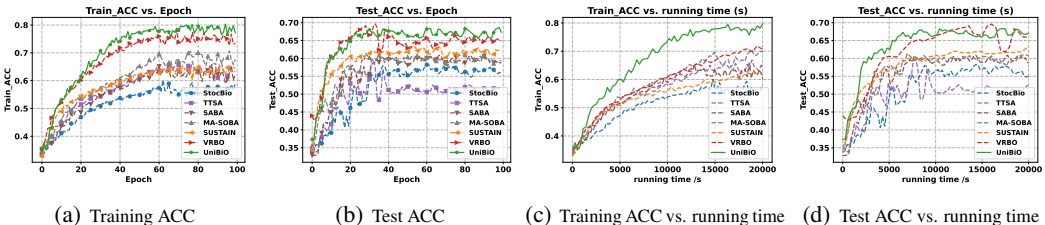

(a) Training ACC     (b) Test ACC     (c) Training ACC vs. running time     (d) Test ACC vs. running time

Figure 4: Results of bilevel optimization on data hyper-cleaning with noise $\tilde{p} = 0.1$ and $p = 4$. Subfigure (a), (b) show the training and test accuracy with the training epoch. Subfigure (c), (d) show the training and test accuracy with the running time.

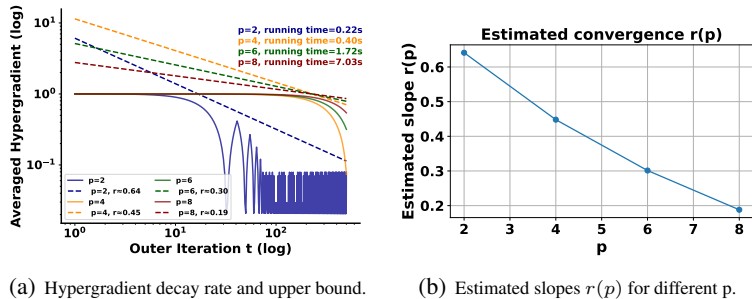

(a) Hypergradient decay rate and upper bound.     (b) Estimated slopes $r(p)$ for different p.

Figure 5: Log–log plot of the convergence behavior of the averaged hypergradient norm under different uniform-convexity parameters $p$.

### E.3 ESTIMATION OF THE CONVERGENCE RATE FOR DIFFERENT $p$

We adopt the same configuration as in the synthetic experiment under deterministic setting (i.e., no gradient noise) with outer iteration $T = 500$ iterations in Algorithm 2. Recall that our theory guarantees a power-law decay of the averaged hypergradient:

$$\frac{1}{t} \sum_{i=1}^{t} \|\nabla \Phi_p(x_i)\| \lesssim t^{-r(p)}.$$

Taking logarithms on both sides yields

$$\log\left(\frac{1}{t} \sum_{i=1}^{t} \|\nabla \Phi_p(x_i)\|\right) \leq -r(p) \log(t) + C,$$

where the slope $-r(p)$ characterizes an upper bound on the convergence rate, and $C$ is a universal constant.

In Figure 5(a), the *solid curve* represents the empirically observed sequence of averaged hypergradient norms, whereas the *dashed curve* corresponds to the fitted power-law upper bound, obtained via a linear regression on the log–log plot. We also report the runtime for different values of $p$.

Figure 5(b) reports the resulting fitted curves and the estimated slopes for $p \in \{2, 4, 6, 8\}$. As $p$ increases, the slope magnitude decreases, indicating slower convergence. This is consistent with our complexity results as shown in Theorem 5.1.

An additional observation is that the empirical convergence rates are *strictly faster* than our theoretical worst-case bound $O(\epsilon^{-3p+2})$ outer iterations required to find an $\epsilon$-stationary point (see Equation (63)). This suggests either that our example is not a hard instance or that the current complexity bound may not be tight; we leave a tighter characterization for future work. Note that there is an extra $\widetilde{O}(\epsilon^{-5p+6})$ inner iterations complexity which is reflected in the runtime result in Figure 5. In particular, the averaged inner iterations for various $p = [2, 4, 6, 8]$ are $[75, 172, 737, 3059]$, which means that larger $p$ significantly increases the inner-loop iterations (i.e., the choice of $K_t$ as chosen in Theorem C.5) used in the subroutine Epoch-SGD (i.e., Algorithm 1).

## F    Hyerparameter Setting

For a fair comparison, we carefully tune the hyperparameters for each baseline, including upper- and lower-level step sizes, the number of inner loops, momentum parameters, etc. For the data hyper-cleaning experiments, the upper-level learning rate $\eta$ and the lower-level learning rate $\gamma$ are selected from range $[0.001, 0.1]$. The best $(\eta, \gamma)$ are summarized as follows: Stocbio: $(0.01, 0.002)$, TTSA: $(0.001, 0.02)$, SABA: $(0.05, 0.02)$, MA-SOBA: $(0.01, 0.01)$, SUSTAIN: $(0.05, 0.05)$, VRBO: $(0.1, 0.05)$, UniBiO: $(0.05, 0.02)$. The number for neumann series estimation in StocBiO and VRBO is fixed to 3, while it is uniformly sampled from $\{1, 2, 3\}$ in TTSA, and SUSTAIN. The batch size is set to be 128 for all algorithms except VRBO, which uses larger batch size of 256 (tuned in the range of $\{63, 128, 256, 512, 1024\}$) at the checkpoint step and 128 otherwise. UniBiO uses the periodic update for low-level variable and sets the iterations $N = 3$ and the update interval $I = 2$. The momentum parameter $\beta$ is fixed to 0.9 in MA-SOBA and UniBiO.

## G    The Use of Large Language Models (LLMs)

LLMs are not involved in our research methodology or analysis. Their use is limited to polish the writing.

