# OpenReview forum: "Bilevel Optimization with Lower-Level Uniform Convexity: Theory and Algorithm"
_ICLR.cc/2026/Conference — ICLR 2026 Poster_

### Official Review · Reviewer_DjaX · 2025-10-26

**Soundness:** 2
**Presentation:** 3
**Contribution:** 2
**Rating:** 6
**Confidence:** 3

**Summary:**

The paper introduces a new tractable problem class for bilevel optimization: lower-level uniform convexity (LLUC), and proposes an algorithm that achieves the $\tilde{\mathcal{O}}(\epsilon^{-5p+6})$ oracle complexity bound for finding an $\epsilon$-stationary point when the lower-level function is uniformly convex with an exponent $p \ge 2$. The result in this paper recovers the known near-optimal rate when $p=2$.

The proposed algorithm consists of two loops: the inner loop applies Epoch-SGD (tailored to uniformly convex problems), and the outer loop applies normalized SGD with momentum to tackle nonconvex optimization under the generalized smooth assumption.

**Strengths:**

I believe the main advantage of this paper lies in its originality, as it presents new categories of problems and studies the solution algorithms under these settings.

Moreover, the proposed algorithms can recover the (near)-optimal complexity for known settings ($p=2$).

The Holder continuity of the lower-level optimal solution mapping (Lemma 4.2) and the implicit function theorem under the LLUC assumption (Lemma 4.3) presented in this paper are of independent interest and may also be helpful for other problems.

**Weaknesses:**

Although I appreciate the effort to introduce new problem classes for bilevel optimization in this paper, the assumptions of the introduced class do not look very clean:

1. In the last item of  Assumption 3.2,  it is assumed that $\lambda_{\min} ( \frac{{\rm d} \nabla_y g(x,y)}{{\rm d} [y]^{\circ (p-1)}} ) \ge \mu >0$. This assumption seems to be derived from the proof; can the authors provide practical examples that satisfy these conditions? For instance, does the application in Section 3.2 meet this assumption?

 2. The second item of Assumption 3.4 looks strange. While the other items all assume bounded variance, the second part assumes that the noise follows a sub-Gaussian distribution. Could this be a typo?

**Questions:**

Please see the weakness part.

---

> ### Author Response · Authors · 2025-11-18
>
> **Q1. In the last item of Assumption 3.2, it is assumed that $\lambda_{\min} ( \frac{{\rm d} \nabla_y g(x,y)}{{\rm d} [y]^{\circ (p-1)}} ) \ge \mu >0$. This assumption seems to be derived from the proof; can the authors provide practical examples that satisfy these conditions? For instance, does the application in Section 3.2 meet this assumption?**
>
> **A1.** We appreciate your question. As noted in the remark, Assumption 3.2(v) can alternatively be replaced by the condition that
> $\frac{d\nabla_y g(x,y)}{d[y]^{\circ (p-1)}}$
> is independent of
> $[y]^{\circ (p-1)}$.
>
> Regarding the examples in Section 3.2, both of them satisfy
> $\lambda_{\min} \left( \frac{ d \nabla_y g(x,y)}{{ d} [y]^{\circ (p-1)}} \right) \ge \mu > 0$.
> In particular, we have
> - **Example 1:**
>   $
>   \lambda_{\min}\left(
>   \frac{d \nabla_y g(x,y)}{{\rm d}[y]^{\circ (p-1)}}
>   \right) = 1 > 0.
>   $
>
> - **Example 2:**
>   $
>   \lambda_{\min}\left(
>   \frac{d \nabla_y g(x,y)}{{\rm d}[y]^{\circ (p-1)}}
>   \right) > p c > 0.
>   $
> Hence, both examples satisfy Assumption 3.2(v) and demonstrate that this condition naturally holds in practical LLUC settings.
> Details can be found in the Appendix A.2.
>
> **Q2. The second item of Assumption 3.4 looks strange. While the other items all assume bounded variance, the second part assumes that the noise follows a sub-Gaussian distribution. Could this be a typo?**
>
> **A2.** This is not a typo, the second part of Assumption 3.4 is intentionally stated in a sub-Gaussian form to enable high probability analysis for the lower-level problem.
>
> As stated in Theorem 5.1, the expectation is taken only over the randomness in
>  $\mathcal{F}_{T+1}$,
> which is independent of $\mathcal{F}_y$. Therefore we can use high probability and expectation simultaneously. In fact, if we replace the other bounded variance assumptions in Assumption 3.4 with sub-Gaussian or bounded noise assumptions, we can obtain a high probability convergence result over all sources of randomness.
>
> The reason of using high probability and expectation together is illustrated as following.
> Our upper-level analysis requires the hypergradient estimator to be sufficiently accurate, which in turn requires the lower-level estimation error
> $\\|y_t-y_t^\*\\|$
> to remain small for all $t$.
>  Assumption 3.2 does not impose the classical smoothness condition and relies only on the uniform convexity and $(L_0,L_1)$-smoothness of the lower-level problem.
> Ensuring
> $\\|y_t-y_t^\*\\|$
> stays small for all $t$ requires the gradient of the lower-level objective to be bounded along the optimization trajectory (under our current analysis). To achieve this, we applied a shrinking-ball technique similar to [1], and generalized it to the $(\mu,p)$-uniformly convex setting, which guarantees with high probability (over the randomness in
> $\mathcal{F}_y$)
> that
> $\\|y_t-y_t^\*\\|$
>  is small for all $t$ and that the gradient of the lower-level objective remains bounded along the trajectory (see Lemma 5.2 and its proof in Appendix C.2).
>
> [1] Hazan, Elad, and Satyen Kale. "Beyond the regret minimization barrier: optimal algorithms for stochastic strongly-convex optimization." The Journal of Machine Learning Research 15.1 (2014): 2489-2512.

---

### Official Review · Reviewer_qtYP · 2025-10-29

**Soundness:** 3
**Presentation:** 1
**Contribution:** 2
**Rating:** 4
**Confidence:** 4

**Summary:**

This paper identifies a tractable class of bilevel optimization problems that interpolates between lower-level strong convexity and general convexity via lower-level uniform convexity (LLUC). For uniformly convex lower-level functions with exponent $p \ge 2$, the authors establish an implicit differentiation theorem characterizing the smoothness of the hyperobjective. Building on this, they propose a stochastic algorithm, termed UniBiO, with provable convergence guarantees, based on an oracle that provides stochastic gradient and Hessian-vector product information. The algorithm achieves an oracle complexity bound of $\tilde{O}(\epsilon^{-5p+6})$ for finding $\epsilon$-stationary points.

**Strengths:**

1. This paper introduces the concept of lower-level uniform convexity (LLUC) to bilevel optimization, which is new.

2. For uniformly convex lower-level functions with exponent $p \ge 2$, the authors establish an implicit differentiation theorem that characterizes the smoothness of the hyperobjective.

3. The authors design a stochastic algorithm, termed UniBiO, with provable convergence guarantees, and their algorithm achieves $\tilde{O}(\epsilon^{-5p+6})$ oracle complexity bound for finding $\epsilon$-stationary points.

**Weaknesses:**

1. My primary concerns relate to the assumptions and presentation of the paper:

1-1. In Assumption 3.2(v), the notation $d[y]^{o~p-1}$ is not clearly given. Although the authors refer to Theorem 3.1, this theorem does not appear in the text. The authors also refer to Theorem A.1, this theorem also does not appear in the text (may be Definition A.1). Such omissions are critical, as this definition is important in the paper.

Moreover, similar questions exist in this paper, e.g., in line 209, it should be Assumption 3.4, rather than Theorem 3.4.

1-2. Some assumptions appear strong. In Assumption 3.2(v), the generalized Jacobian is assumed to satisfy $\lambda_{\min}\left(\frac{d\nabla g(x,y)}{d[y]^{o\~p-1}}\right) > 0$. Does this imply that $\frac{d\nabla g(x,y)}{d[y]^{o\~p-1}}$ is invertible at $y^{o\~p-1}$? If so, Theorem 4.1 may simply review standard results for hypergradients [Ghadimi \& Wang, 2018], rather than offering new insights.

1-3. The definitions of $F$ and $G$ are not given in Assumption 3.4. Moreover, the paper focuses on stochastic methods for bilevel optimization, but there is insufficient discussion of stochastic settings.

1-4. This paper is not well-written, which will make the reader confused. For example, in Sections 4.1 and 5.3, the authors introduce the proof sketches of some theorems. However, these statements can be moved to the appendix. Conversely, some important elements, such as the stochastic approximate hypergradient $\hat{\nabla}f(x_t, y_t;\bar{\xi}_t)$ in Algorithm 2, are only introduced in the appendix and should be moved to the main text.

**Questions:**

1. What is the difference between $[y]^{o\~p-1}$ and $y^{o\~p-1}$?

2. In Assumption 3.3(iv), is the constant $\Delta_{\Phi}$ known? If not, would it be more appropriate to assume that $\Phi$ is lower-bounded?


3. In Lemma 4.3, would it be much better to analyze the error between the stochastic approximate hypergradient $\hat{\nabla}f(x_t, y_t;\bar{\xi}_t)$ and $\nabla\Phi(x_t)$, rather than $\widehat{\nabla}\Phi(x_t)$? Since the former is used in Algorithm 2 and $\widehat{\nabla}\Phi(x_t)$ is just an auxiliary quantity.

4. In Theorem 5.1, the definitions of $I$ and $Q$ are not provided, and the detailed form of $\delta$ is omitted. Moreover, standard convergence results for stochastic bilevel optimization [Ghadimi \& Wang, 2018; Ji et al., 2021; Chen et al., 2021] typically state $\frac{1}{T}\sum_{i=1}^T \mathbb{E}\\|\nabla\Phi(x_t)\\| \le \epsilon$ in expectation, not with high probability. Can the authors clarify why high probability and expectation are used simultaneously in their result?

---

> ### Author Response · Authors · 2025-11-18
>
> **Q1.  In Assumption 3.2(v), the notation $d[y]^{\circ~p-1}$ is not clearly given. Although the authors refer to Theorem 3.1, this theorem does not appear in the text (may be Definition A.1). Such omissions are critical, as this definition is important in the paper.
>  is not clearly given.
> Moreover, similar questions exist in this paper, e.g. in line 209, it should be Assumption 3.4, rather than Theorem 3.4.**
>
> **A1.** We appreciate your careful reading of our paper. The definition of ${{ d}[y]^{\circ (p-1)}}$ has been clarified in the Assumption 3.2 (marked in red). Specifically, it represents a change-of-variable differential, introduced to simplify the analysis under the transformation $u = y^{\circ (p-1)}$.
>
> **Q2. Some assumptions appear strong. In Assumption 3.2(v), the generalized Jacobian is assumed to satisfy $\lambda_{\min}\left(\frac{d\nabla_y g(x,y)}{d[y]^{\circ p-1}}\right) > 0$. Does this imply that $\frac{d\nabla_y g(x,y)}{d[y]^{\circ p-1}}$ is invertible at $y^{\circ~p-1}$? If so, Theorem 4.1 may simply review standard results for hypergradients [1], rather than offering new insights.**
>
>
> **A2.** Assumption3.2(v) indeed ensures that
> $
> \frac{{\rm d}\nabla_y g(x,y)}{{\rm d}[y]^{\circ(p-1)}}$
> is invertible. However, as clarified in the remark, this can be equivalently replaced by another condition that
> $
> \frac{{\rm d}\nabla_y g(x,y)}{{\rm d}[y]^{\circ(p-1)}}$
> is independent of $[y]^{\circ(p-1)}$ (line 181--182).
>
> Lemma B.2 shows that this independence assumption, together with uniform convexity, already implies invertibility via our generalized implicit function theorem without directly assuming positive definiteness of the generalized Hessian. The two conditions are not equivalent; our result therefore extends standard bilevel analyses such as [1] under the LLUC framework.
>
> **Q3.  The definitions of $F$ and $G$ are not given in Assumption 3.4. Moreover, the paper focuses on stochastic methods for bilevel optimization, but there is insufficient discussion of stochastic settings.**
>
> **A3.** Thank you for pointing this out. We would like to clarify that the definitions of
> $F,G$ are already provided in line 140.
>
> **Q4. This paper is not well-written, which will make the reader confused. For example, in Sections 4.1 and 5.3, the authors introduce the proof sketches of some theorems. However, these statements can be moved to the appendix. Conversely, some important elements, such as the stochastic approximate hypergradient
> $\hat{\nabla}f(x_t, y_t;\bar{\xi}_t)$
> in Algorithm 2 are only introduced in the appendix and should be moved to the main text .**
>
> **A4.**  Thanks for your suggestion. We have reorganized the proof and also moved the stochastic approximate hypergradient to the main text.
>
> **Q5. What is the difference between $[y]^{\circ p-1}$ and $y^{\circ p-1}$?**
>
> **A5.** They are the same. We have fixed this typo in the revised version.
>
> **Q6. In Assumption 3.3(iv), is the constant $\Delta_{\Phi}$ known? If not, would it be more appropriate to assume that $\Phi$ is lower-bounded?**
>
> **A6.** Yes, the constant $\Delta_{\Phi}$ is typically known in machine learning. In typical machine
> learning settings, the loss function $\Phi$ is nonnegative and therefore
> lower-bounded by $0$. Consequently,
> $
> \Delta_{\Phi}
> = \Phi(x_0) - \inf_x \Phi(x)
> \le \Phi(x_0),
> $
> and $\Phi(x_0)$ is known at initialization. We can always use
> $\Phi(x_0)$ as a valid upper bound for $\Delta_{\Phi}$.
>
> **Q7. In Lemma 4.3, would it be much better to analyze the error between the stochastic approximate hypergradient $\hat{\nabla}f(x_t, y_t;\bar{\xi}_t)$ and $\nabla\Phi(x_t)$, rather than $\widehat{\nabla}\Phi(x_t)$? Since the former is used in Algorithm 2 and $\widehat{\nabla}\Phi(x_t)$ is just an auxiliary quantity.**
>
> **A7.** Lemma 4.3 is primarily intended to illustrate the bias introduced by the inexact hypergradient oracle. We agree that integrating its analysis with the results improves clarity, and we have accordingly consolidated the related content into Appendix B in the revised version.
>
>
>
>
> [1] Ghadimi, Saeed, and Mengdi Wang. "Approximation methods for bilevel programming." arXiv preprint arXiv:1802.02246 (2018).

---

> ### Author Response · Authors · 2025-11-18
>
> **Q8. In Theorem 5.1, the definitions of $I$ and $Q$ are not provided, and the detailed form of $\delta$ is omitted. Moreover, standard convergence results for stochastic bilevel optimization [Ghadimi \& Wang, 2018; Ji et al., 2021; Chen et al., 2021] typically state $\frac{1}{T}\sum_{i=1}^T \mathbb{E}\\|\nabla\Phi(x_t)\\| \le \epsilon$ in expectation, not with high probability. Can the authors clarify why high probability and expectation are used simultaneously in their result?**
>
> **A8.** $I$ is the lower-level update frequency introduced in Algorithm 2, and $Q$ is the batch size required by the Neumann series approach to approximate the hypergradient (see equation (4)).
> The parameter $\delta$ denotes the failure probability commonly used in high-probability analysis.
>
> As stated in Theorem 5.1, the expectation is taken only over the randomness in $\mathcal{F}_{T+1}$, which is independent of $\mathcal{F}_y$. Therefore we can use high probability and expectation simultaneously. In fact, if we replace the other bounded variance assumptions in Assumption 3.4 with sub-Gaussian or bounded noise assumptions, we can obtain a high probability convergence result over all sources of randomness.
>
> The reason of using high probability and expectation together is illustrated as following. Our upper-level analysis requires the hypergradient estimator to be sufficiently accurate, which in turn requires the lower-level estimation error
> $\\|y_t-y_t\^*\\|$ to remain small for all $t$. Assumption 3.2 does not impose the classical smoothness condition and relies only on the uniform convexity and $(L_0,L_1)$-smoothness of the lower-level problem. Ensuring
>  $\\|y_t-y_t^\*\\|$
> stays small for all $t$ requires the gradient of the lower-level objective to be bounded along the optimization trajectory (under our current analysis). To achieve this, we applied a shrinking-ball technique similar to [1], and generalized it to the $(\mu,p)$-uniformly convex setting, which guarantees with high probability (over the randomness in $\mathcal{F}_y$) that
> $\\|y_t-y_t^\*\\|$ is small for all $t$ and that the gradient of the lower-level objective remains bounded along the trajectory (see Lemma 5.2 and its proof in Appendix C.2).
>
> [1] Hazan, Elad, and Satyen Kale. "Beyond the regret minimization barrier: optimal algorithms for stochastic strongly-convex optimization." The Journal of Machine Learning Research 15.1 (2014): 2489-2512.

---

> > ### Comment · Reviewer_qtYP · 2025-11-26
> > **Thanks for the authors' responses.**
> >
> > The author's response addressed many of my concerns, and I have no further questions. However, note that the presentation of this paper should be improved in the revision. I will raise the rating to 6.

---

> > > ### Author Response · Authors · 2025-11-26
> > > **Thank you for your review**
> > >
> > > Dear Reviewer qtYP,
> > >
> > > We sincerely appreciate your constructive comments. We are glad that our clarifications resolved the concerns you raised. The manuscript has been revised and updated, and all modifications are marked in red. Please feel free to share any further feedback. We will ensure additional polishing in the final version of this paper.
> > >
> > > Best,
> > >
> > > Authors

---

### Official Review · Reviewer_pm1j · 2025-10-31

**Soundness:** 3
**Presentation:** 3
**Contribution:** 3
**Rating:** 6
**Confidence:** 3

**Summary:**

This paper addresses the challenge of bilevel optimization when the lower-level problem is not strongly convex. The authors identify a tractable problem class characterized by uniform convexity in the lower-level function. They establish a new implicit differentiation theorem for this class and propose a stochastic algorithm, UniBio, with a convergence guarantee of $\widetilde{O}(\epsilon^{-5p+6})$ for finding an $\epsilon$-stationary point. Notably, this rate matches the optimal complexity for the strongly convex case ($p=2$). The theoretical findings are supported by experiments on synthetic and data hyper-cleaning tasks.

**Strengths:**

1. The paper identifies a novel and tractable class of bilevel problems with uniformly convex lower-level functions, providing a crucial pathway between strong convexity and general convexity.

2. The presentation is clear and well structured.

**Weaknesses:**

The oracle complexity $\widetilde{O}(\epsilon^{-5p+6})$ becomes prohibitively high for large $p$, creating a significant gap between theoretical tractability and practical efficiency for near-general convex problems.

**Questions:**

1. In Algorithm 2 (line 8), a momentum term is utilized in the iterative update. What is the precise role of this term in the convergence analysis? Would its removal deteriorate the final oracle complexity?

2. As noted in the weaknesses, the practical efficiency for larger values of the uniform convexity exponent `p` remains a concern. Could the authors include additional experiments, perhaps on a synthetic task, to explicitly investigate and demonstrate how the value of `p` influences the convergence speed and runtime of the proposed UniBio algorithm?

3. The analysis heavily relies on the uniform convexity of the lower-level problem. Is there a potential extension of these results under a generalized Polyak-Lojasiewicz (PL) condition for the lower-level function? Could a similar tractable class and convergence guarantee be established under such a condition?

4. A minor suggestion: please check some typos, such as “theoremthat” on page2; “$T_{k}$” in line 7 in Alg1 on page7.

---

> ### Author Response · Authors · 2025-11-18
>
> **Q1. The oracle complexity $\widetilde{O}(\epsilon^{-5p+6})$ becomes prohibitively high for large $p$, creating a significant gap between theoretical tractability and practical efficiency for near-general convex problems.
>  $\widetilde{O}(\epsilon^{-5p+6})$ becomes prohibitively high for large $p$, creating a significant gap between theoretical tractability and practical efficiency for near-general convex problems.**
>
> **A1.** We appreciate your observation. When the lower-level problem is strongly convex (i.e., $p=2$), our framework achieves the optimal oracle complexity $\widetilde{O}(\epsilon^{-4})$, matching the state-of-the-art convergence rates without mean-squared smoothness assumption of stochastic estimators [1]. In contrast, when the lower-level problem is merely convex (i.e., $p \to +\infty$), finding small hypergradient is intractable in general [2]. We have mentioned it in our abstract (see line 015).
>
> Thus, our complexity bound is tight up to logarithmic factors for both $p=2$ and $p\to +\infty$. We acknowledge that the bound for intermediate values $p\in(2,+\infty)$ may not be optimal, and we plan to investigate this in future work.
>
> **Q2. In Algorithm 2 (line 8), a momentum term is utilized in the iterative update. What is the precise role of this term in the convergence analysis? Would its removal deteriorate the final oracle complexity?**
>
> **A2.** The momentum term can provably eliminate the need for large batch sizes on nonconvex objectives [3]. Removing the momentum update does not change the final oracle complexity, but it does require a large batch size to reduce the variance in hypergradient estimation, which is generally impractical. For example, the work of [4] does not leverage momentum term and hence requires large minibatch size in their analysis (e.g., Theorem 4 in [4])
>
>
> [1] Chen, Xuxing, Tesi Xiao, and Krishnakumar Balasubramanian. "Optimal algorithms for stochastic bilevel optimization under relaxed smoothness conditions." Journal of Machine Learning Research 25.151 (2024): 1-51.
>
> [2] Lesi Chen, Jing Xu, and Jingzhao Zhang. "On finding small hyper-gradients in bilevel optimization: Hardness results and improved analysis." In The Thirty Seventh Annual Conference on Learning Theory, pp. 947-980. PMLR, 2024.
>
> [3] Cutkosky, Ashok, and Harsh Mehta. "Momentum improves normalized sgd." International conference on machine learning. PMLR, 2020.
>
> [4] Kaiyi Ji, Junjie Yang, and Yingbin Liang. "Bilevel Optimization: Convergence Analysis and Enhanced Design." In International Conference on Machine Learning, pp. 4882-4892. PMLR, 2021.

---

> > ### Author Response · Authors · 2025-11-18
> >
> > **Q3. As noted in the weaknesses, the practical efficiency for larger values of the uniform convexity exponent $p$ remains a concern. Could the authors include additional experiments, perhaps on a synthetic task, to explicitly investigate and demonstrate how the value of $p$ influences the convergence speed and runtime of the proposed UniBio algorithm?**
> >
> > **A3.** Our synthetic experiment illustrates how the lower-level curvature parameter $p$ affects the computational complexity of bilevel optimization.
> > As $p$ increases, the convergence speed of the hypergradient slows down, which is consistent with our theory. We have added an figure in Section E.3 in Appendix to characterize how the convergence speed varies over $p$.
> >
> > **Q4. The analysis heavily relies on the uniform convexity of the lower-level problem. Is there a potential extension of these results under a generalized Polyak-Lojasiewicz (PL) condition for the lower-level function? Could a similar tractable class and convergence guarantee be established under such a condition?**
> >
> > **A4.** We appreciate your question regarding an extension under the generalized PL condition.
> >
> > In the convex case, the PL condition indeed implies strong convexity [1]
> > , which is the standard assumption adopted in prior bilevel optimization analysis [2] [3] [4]. Our framework strictly generalizes this setting by allowing the lower-level problem to satisfy a broader form of uniform convexity beyond the strongly convex  case.
> >
> > In the nonconvex case, our work is orthogonal to the PL condition. Studies addressing nonconvex PL lower-level problems (e.g. [5]) require additional technical assumptions such as solution-set Lipschitz continuity (Assumption 5 in [5]) and the Linear Independence Constraint Qualification (LICQ) (Assumption 6 in [5]) to ensure that the hypergradient is well-defined. In contrast, our framework does not rely on  solution-set Lipschitz continuity or LICQ, and also cannot handle nonconvex lower-level problems since lower-level uniform convexity requires the lower-level function to be convex.
> >
> >
> > In summary, our analysis characterizes the curvature of the lower-level function through its $p$-order growth property, which provides a uniform convexity structure fundamentally different from the PL condition. For this reason, an extension of our results to the generalized PL setting does not appear feasible.
> >
> > **Q5. A minor suggestion: please check some typos, such as "theoremthat" on page2; $T_{k}$ in line 7 in Alg1 on page7.**
> >
> > **A5.** Thanks for your suggestion. We have fixed them in the revised version.
> >
> > [1] Karimi, Hamed, Julie Nutini, and Mark Schmidt. "Linear convergence of gradient and proximal-gradient methods under the polyak-łojasiewicz condition." Joint European conference on machine learning and knowledge discovery in databases. Cham: Springer International Publishing, 2016.
> >
> > [2] Ghadimi, Saeed, and Mengdi Wang. "Approximation methods for bilevel programming." arXiv preprint arXiv:1802.02246 (2018).
> >
> > [3] Kaiyi Ji, Junjie Yang, and Yingbin Liang. "Bilevel Optimization: Convergence Analysis and Enhanced Design." In International Conference on Machine Learning, pp. 4882-4892. PMLR, 2021.
> >
> > [4] Hong, Mingyi, et al. "A two-timescale stochastic algorithm framework for bilevel optimization: Complexity analysis and application to actor-critic." SIAM Journal on Optimization 33.1 (2023): 147-180.
> >
> > [5] Kwon, Jeongyeol, et al. "On penalty methods for nonconvex bilevel optimization and first-order stochastic approximation." arXiv preprint arXiv:2309.01753 (2023).

---

> > > ### Comment · Reviewer_pm1j · 2025-11-27
> > >
> > > The author's response have addressed my concerns, and I have no further questions.  I will raise the rating to 8.

---

> > > > ### Author Response · Authors · 2025-11-27
> > > > **Thank you for your review.**
> > > >
> > > > Dear Reviewer pm1j,
> > > >
> > > > We appreciate your thoughtful review of our paper. We are glad that our responses addressed your concerns.
> > > >
> > > > Best,
> > > >
> > > > Authors

---

### Official Review · Reviewer_s2ox · 2025-10-31

**Soundness:** 2
**Presentation:** 2
**Contribution:** 2
**Rating:** 4
**Confidence:** 3

**Summary:**

This paper introduces the concept of lower-level uniform convexity (LLUC) as an intermediate class between lower-level strong convexity (LLSC) and general convexity in bilevel optimization. Under LLUC (parameterized by exponent $p \geq 2$), the authors (i) develop an implicit differentiation theorem characterizing the smoothness of the hyperobjective (showing that the hypergradient exists and is Hölder continuous with order depending on $p$), and (ii) propose **UniBiO**, a stochastic algorithm using normalized momentum for the upper level and a multistage Epoch-SGD with a shrinking-ball strategy for the lower level. They prove an oracle complexity of $\tilde{O}(\varepsilon^{-5p + 6})$ to find an $\varepsilon$-stationary point, which reduces to the known optimal complexity when $p = 2$. Experiments on synthetic data and a data hypercleaning task are provided. The work aims to formalize a tractable intermediate class between LLSC and general convexity and provide the first theoretical and algorithmic results for this class.

**Strengths:**

1. The paper addresses an important and timely problem by extending bilevel optimization analysis beyond the conventional lower-level strong convexity (LLSC) assumption.
2. The introduction of lower-level uniform convexity (LLUC) as an intermediate class is a novel theoretical contribution.
3. A new implicit differentiation theorem is derived for the LLUC setting.

**Weaknesses:**

1. The practical motivation for LLUC is not sufficiently justified; the provided examples (e.g., $\ell_p$-regression) appear contrived and do not reflect modern, complex bilevel learning tasks.
2. The theoretical framework relies on multiple technical and non-standard assumptions to establish convergence.
3. The proposed UniBiO algorithm appears structurally similar to existing methods (e.g., BO-REP), with limited algorithmic innovation.

**Questions:**

1. Can the authors provide a compelling, real-world machine learning example where the lower-level problem is uniformly convex with $p > 2$ but not strongly convex ($p = 2$)?
2. What is the core algorithmic innovation in UniBiO compared to existing bilevel optimization frameworks such as BO-REP?

---

> ### Author Response · Authors · 2025-11-18
>
> **Q1. The practical motivation for LLUC is not sufficiently justified; the provided examples (e.g., $\ell_p$-regression) appear contrived and do not reflect modern, complex bilevel learning tasks.**
>
> **A1.**  This is not a contrived bilevel problem. For example, in the literature, $\ell_p$-hyperparameter learning can be formulated as a bilevel optimization problem [1]. The mathematical formulation is
>
> $\min_{\lambda \ge 0}  f(w^*(\lambda)),$
>
> s.t.
> $w^*(\lambda) = \arg\min_{w \in \mathbb{R}^n}\left( g(w) + \lambda R(w) \right),$
>
> where $f : \mathbb{R}^n \to \mathbb{R}$ is continuously differentiable, $\lambda\ge 0$ is the regularization strength, $g$ is convex, and the regularizer is
> $
> R(w) := \\|w\\|_p^p
> $
> with $ p\ge 2.$
> The lower-level problem has a uniformly convex regularizer with exponent $p$.
> A concrete instance of this framework is the linear-regression bilevel problem with
> $\ell_p$ hyperparameter (e.g., [1], Eq.~(33) with $p\geq 2$):
>
> $
> \min_{\lambda}
> \\|A_{\mathrm{val}} w^*(\lambda) - b_{\mathrm{val}}\\|_2^2
> $
>
> s.t.
> $
> w^*(\lambda) = \arg\min_{\hat{w}}
> \left(\\|A_{\mathrm{tr}}\hat{w} - b_{\mathrm{tr}}\\|_2^2+ \lambda\\|\hat{w}\\|_p^p\right).
> $
>
> Here, $w$ is the model parameter, $\lambda$ is the regularization strength, $A_{\mathrm{val}}$ and $b_{\mathrm{val}}$ denote the validation data matrix and labels respectively, and $A_{\mathrm{tr}}$ and $b_{\mathrm{tr}}$ denote the training data matrix and labels.
>
> **We can also extend this formulation to deep learning.**
> Let the network parameters be $w = (u,v)$, where $u$ are the parameters of the
> first $L-1$
> layers (feature extractor) and $v$ is the last-layer classifier.
> Given a feature map $\phi_u(x)$ induced by $u$, consider a convex loss
> $\ell(\cdot,y)$ (e.g., squared or logistic loss) and define
>
> $
> \mathcal{L}\_{\mathrm{tr}}(u,v)
> := \frac{1}{N_{\mathrm{tr}}} \sum_{i=1}^{N_{\mathrm{tr}}}
> \ell\big(\langle \phi_u(x_i), v\rangle, y_i\big),
> $
>
> $
> \mathcal{L}\_{\mathrm{val}}(u,v)
> :=\frac{1}{N_{\mathrm{val}}}\sum_{i=1}^{N_{\mathrm{val}}}
> \ell\big(\langle \phi_u(x_i^{\mathrm{val}}), v\rangle, y_i^{\mathrm{val}}\big).
> $
>
> We can consider the following bilevel problem:
>
> $
> \min_{u,\lambda \ge 0}
> \mathcal{L}_{\mathrm{val}}\big(u, v^*(u,\lambda)\big),
> $
>
> s.t.
> $
> v^*(u,\lambda)
> =\arg\min_{v}
> \Big(
> \mathcal{L}_{\mathrm{tr}}(u,v)
> +\lambda \\|v\\|_p^p
> \Big),
> \quad p \ge 2.
> $
>
> For any fixed $(u,\lambda)$ with $\lambda>0$, the function
> $v \mapsto \mathcal{L}_{\mathrm{tr}}(u,v)$ is convex (by convexity of
> $\ell$ in its first argument and linearity in $v$), and the regularizer
> $v \mapsto \lambda\\|v\\|_p^p$ is uniformly convex for $p \ge 2$. Hence the
> lower-level objective is uniformly convex in $v$ and admits a unique
> minimizer $v^*(u,\lambda)$, while the upper level updates both the
> representation $u$ and the hyperparameter $\lambda$ based on validation
> performance.
>
> This pattern appears widely in deep learning, especially in
> transfer learning and fine-tuning: a pretrained feature extractor
> $u$ is refined using validation feedback, while the last layer $v$ is repeatedly
> retrained or solved in closed form for each candidate representation.
> Thus, the example is both practically relevant and theoretically convenient, as
> the uniformly convex lower level enables stable bilevel analysis and implicit
> differentiation.
>
> **Q2. The theoretical framework relies on multiple technical and non-standard assumptions to establish convergence.**
>
> **A2.** We respectfully disagree. Our framework does not require either smoothness or strong convexity of the lower-level problem when $p > 2$, unlike standard bilevel optimization settings [2]. When $p = 2$, our assumptions align fully with [2] and introduce no additional restrictions. Thus, our framework generalizes the standard assumptions to the uniformly convex case rather than relying on stronger or non-standard conditions.
>
> [1] Okuno, Takayuki, et al. "On lp-hyperparameter learning via bilevel nonsmooth optimization." Journal of Machine Learning Research 22.245 (2021): 1-47.
>
> [2] Ghadimi, Saeed, and Mengdi Wang. "Approximation methods for bilevel programming." arXiv preprint arXiv:1802.02246 (2018).

---

> > ### Author Response · Authors · 2025-11-18
> >
> > **Q3. The proposed UniBiO algorithm appears structurally similar to existing methods (e.g., BO-REP), with limited algorithmic innovation.**
> >
> > **A3.** Our paper is significantly different from BO-REP. First, unlike BO-REP which relies on standard implicit differentiation theorem under lower-level strong convexity, our paper derives a new implicit differentiation theorem under lower-level uniform convexity (Theorem 4.1), which makes sure the hypergradient exists even without lower-level strong convexity. This is a prerequisite for deriving efficient bilevel algorithms.
> >
> > Second, the algorithm we proposed is not an easy extension of BO-REP although they may appear conceptually similar. Their lower-level problem is smooth and strongly convex, but we do not have either smoothness or strong convexity for the lower-level problem. Therefore, the theoretical proofs are very different from the proof in BO-REP. For instance, we need to carefully leverage the shrinking ball approach to control the lower-level error without lower-level smoothness and strong convexity (e.g., Lemma C.8 and C.9), which does not appear in BO-REP. Our parameter choices (e.g., learning rate, periodic update interval size, and the number of iterations) are also significantly different from BO-REP.
> >
> > **Q4. Can the authors provide a compelling, real-world machine learning example where the lower-level problem is uniformly convex with $p>2$ but not strongly convex ($p=2$)?**
> >
> > **A4.** We provided real-world examples in the answer A1, including hyperparameter optimization in both linear regression and neural network in the presence of a uniformly convex regularizer.
> >
> > **Q5. What is the core algorithmic innovation in UniBiO compared to existing bilevel optimization frameworks such as BO-REP?**
> >
> > **A5.** Please refer to A3.

---

> ### Author Response · Authors · 2025-11-27
> **Looking forward to post-rebuttal feedback**
>
> Dear Reviewer s2ox,
>
> Thank you for reviewing our paper. We have carefully addressed your concerns by providing real-world examples and clarifications of our algorithm UniBiO's innovations compared with BO-REP.
>
> Please let us know if our responses resolve your questions. If so, we would appreciate your consideration in updating the paper’s rating. Thank you again for your time, and we would be happy to discuss any further issues.
>
> Best,
>
> Authors

---

### Author Response · Authors · 2025-11-29
**Summary of Changes during Rebuttal Period**

Dear AC and Reviewers,

Since the scores are being reverted to their pre-discussion state and no further reviewer discussions or public comments are allowed, we would like to provide a brief summary of the changes we made and how reviewers responded.

All reviewers' concerns are addressed. In our response, we clarified the practical relevance of LLUC with concrete real-world machine learning examples and highlighted the key differences between UniBiO and prior methods such as BO-REP. We further explained the role of momentum, the necessity of high-probability bounds for controlling lower-level error, and clarified our oracle complexity results.

In the revised version of the paper, we reorganized the presentation for clarity, and added new synthetic experiments illustrating how the curvature parameter $p$ affects convergence rate in Appendix E.3. All changes are marked in red in the revised version of this paper.

These revisions collectively resolved the reviewers’ questions. **Reviewer qtYP increased the score from 4 → 6, and Reviewer pm1j increased the score from 6 → 8**. The AC may verify these changes directly in the reviewer comments prior to the score reversion. Reviewer s2ox and Reviewer DjaX have not yet replied, but we believe our responses adequately addressed their concerns as well.

Hopefully this summary of changes can help AC make their decision. Please let us know if you have further questions.

Thank you for handling our paper.

Best,

Authors

---

### Meta-Review · Area_Chair_5Y3o · 2026-01-05

**Summary:**

This paper tackles bilevel optimization under lower-level uniform convexity (LLUC), an assumption bridging lower-level strong convexity (p = 2) and general convexity. The main contribution is a new implicit differentiation / hypergradient regularity theory beyond the lower-level strongly convex setting, together with the UniBiO algorithm and corresponding non-asymptotic complexity guarantees, recovering near-optimal rates at p = 2 and degrading as p increases.

**General sentiment is overall positive**. Reviewers appreciated the technical depth and the value of extending bilevel theory beyond the standard lower-level strongly convex setting.

Key concerns and how they were addressed:

- Practical motivation of LLUC (s2ox): A key concern was whether the LLUC regime is practically motivated rather than a stylized theoretical setting. The rebuttal partially strengthens this point by placing LLUC within a bilevel hyperparameter-learning template (validation loss with a lower-level ERM regularized by $\lambda\|w\|_p^p$).  However, the motivation remains somewhat stylized: the rebuttal does not clearly explain why one would choose $p>2$ regularization in practice (as opposed to the more common $p=2$/adding an $L_2$ term to recover strong convexity), and it does not provide concrete real-world case studies showing uniformly convex-but-not-strongly-convex lower-level objectives. Moreover, the cited reference [1] focuses on the nonsmooth sparse regime $0<p\le 1$, which is not directly aligned with the LLUC setting $p\ge 2$. Overall, despite the motivation being not fully settled, multiple reviewers (DjaX, pm1, qtYP) explicitly recognize LLUC as a new problem class and regard the paper’s results as a theoretical contribution.



- Similarity to BO-REP-style algorithms (s2ox): A central concern was the apparent similarity to BO-REP-style algorithms; however, the authors’ response is helpful in clarifying that the resemblance is largely structural. The paper addresses a different and more challenging regime and derives a new implicit differentiation / hypergradient regularity theory that is not covered by the BO-REP analysis. The final submission should highlight these setting-specific obstacles and explain how the proposed theory overcomes them, making the contribution unmistakable relative to BO-REP.Clarity of assumptions and probabilistic setup (s2ox, pm1j, qtYP, DjaX): Reviewers also raised clarity questions about assumptions and the probabilistic setup. The authors’ revisions and explanations improve readability, although the camera-ready should further polish notation and explicitly map assumptions to the results. The final submission should carefully check for any remaining inconsistencies or errors, and explicitly map each assumption to the corresponding results to further strengthen clarity.

At least two reviewers explicitly indicated they would raise their scores after the rebuttal (qtYP: 4→6; pm1j: 6→8), while the remaining reviewers were more cautious but not fundamentally opposed.

**Ref**:

[1] Okuno, Takayuki, et al., On $l_p$-hyperparameter learning via bilevel nonsmooth optimization, JMLR 2021.

**Reviewer Concerns:**

**Addressed:**

   LLUC realism (s2ox):
  The reviewer questioned whether LLUC is a realistic modeling assumption. The rebuttal provides clearer ML-motivated instances (including a deep-learning formulation) where the lower-level objective can be uniformly convex.

Clarity: definitions, notation, probabilistic setup, assumptions (qtYP, DjaX, s2ox):
  These reviewers flagged missing definitions, notation inconsistencies, unclear use of expectation vs. high-probability statements, and potentially strong assumptions. The rebuttal addresses these points by adding missing definitions, clarifying the probabilistic setting, and providing further explanation/examples for key assumptions. This improves clarity substantially.

Momentum, complexity/efficiency, practicality as p increases (pm1j):
  The rebuttal explains the design rationale (including the role of momentum) and clarifies how the inner-loop schedule and overall oracle complexity scale with *p*, supplemented by additional empirical evidence (including synthetic results) illustrating performance trends as *p* varies. This addresses the main practicality/complexity concern at a reasonable level for the current submission.

Novelty beyond BO-REP and related frameworks (s2ox):
  The rebuttal clarifies that, while the overall bilevel structure may look superficially similar, the main contribution is the development of an implicit differentiation / hypergradient regularity theory under the LLUC regime (beyond the standard lower-level strongly convex and smooth setting). Establishing differentiability and regularity of the hyperobjective in this more challenging regime requires materially different arguments and yields the corresponding complexity guarantees for UniBiO. This addresses the main novelty concern.

**Outstanding:**

Notation/assumption polish and mapping to results (qtYP):   While improved, the final version would still benefit from careful checking and polishing, especially around assumption statements and notation consistency, and by explicitly mapping each assumption to the specific lemmas/theorems that use it.

**Reviewer Scores:**

**Reviewer s2ox (Score: 4 -> Est. unchanged):**

  Unresponsive during the discussion phase. The score would likely remain unchanged. The rebuttal clarifies the main novelty (implicit differentiation / hypergradient regularity theory under LLUC) and improves positioning relative to BO-REP, but lingering skepticism about the strength/checkability of assumptions would likely keep the score similar.

**Reviewer pm1j (Score: 6 -> Est. 8):**

  The reviewer indicated a score increase from **6 → 8** after the rebuttal. With full discussion, the score would likely stay around **8**, since the rebuttal addresses the momentum rationale and clarifies how inner-loop scheduling and overall oracle complexity scale with *p*, supported by additional empirical evidence.

**Reviewer qtYP (Score: 4 -> Est. 6):**

  The reviewer indicated a score increase from **4 → 6** after the rebuttal. With full participation, the score would likely remain around **6**, as the main concerns were missing definitions and presentation issues that were largely addressed. The final assessment would depend on careful camera-ready polishing of notation and organization.

**Reviewer DjaX (Score: 6 -> Est. slight increase or unchanged):**

  The authors directly answered the two technical questions by providing concrete examples satisfying Assumption 3.2(v) and clarifying why the sub-Gaussian condition (and the expectation vs. high-probability setup) is intentional for the analysis, which should reduce the reviewer’s uncertainty.

---

### Decision · Program_Chairs · 2026-01-26

Accept (Poster)